# HDAC9-mediated epithelial cell cycle arrest in G2/M contributes to kidney fibrosis in male mice

Yang Zhang [1], Yujie Yang[1], Fan Yang[2], Xiaohan Liu[1], Ping Zhan[1], Jichao Wu[1], Xiaojie Wang[1], Ziying Wang[1], Wei Tang [1], Yu Sun[1], Yan Zhang [1], Qianqian Xu[3], Jin Shang[4], Junhui Zhen[5], Min Liu [1] ✉ & Fan Yi [1] ✉

Renal tubular epithelial cells (TECs) play a key role in kidney fibrosis by mediating cycle arrest at G2/M. However, the key HDAC isoforms and the underlying mechanism that are involved in G2/M arrest of TECs remain unclear. Here, we find that Hdac9 expression is significantly induced in the mouse fibrotic kidneys, especially in proximal tubules, induced by aristolochic acid nephropathy (AAN) or unilateral ureter obstruction (UUO). Tubule-specific deletion of *HDAC9* or pharmacological inhibition by TMP195 attenuates epithelial cell cycle arrest in G2/M, then reduces production of profibrotic cytokine and alleviates tubulointerstitial fibrosis in male mice. In vitro, knockdown or inhibition of *HDAC9* alleviates the loss of epithelial phenotype in TECs and attenuates fibroblasts activation through inhibiting epithelial cell cycle arrest in G2/M. Mechanistically, HDAC9 deacetylates STAT1 and promotes its reactivation, followed by inducing G2/M arrest of TECs, finally leading to tubulointerstitial fibrosis. Collectively, our studies indicate that HDAC9 may be an attractive therapeutic target for kidney fibrosis.

Kidney tubulointerstitial fibrosis, considered as the ultimate common pathway for the development of chronic kidney disease (CKD), involves multiple cell types[1,2]. Tubular epithelial cells (TECs), the major component of the kidney and the important target in progression of AKI-to-chronic kidney disease (CKD) transition, possess a limited repair capacity[3–5]. When the injury is mild, residual TECs proliferate and reconstitute the tubular structure, promoting to successful remodeling/repair[3]. Otherwise, seriously damaged TECs may undergo changes in energy metabolism, cellular senescence, cell cycle progression, secretion of pro-inflammatory and profibrotic cytokines, or partial epithelial-mesenchymal transition (EMT), leading to maladaptive repair and kidney fibrosis[5]. Remarkably, accumulating evidence shows that prolonged G2/M cell cycle arrest in proximal tubular epithelial cells (PTECs) is an important driver of maladaptive repair and kidney fibrosis[2,6]. PTECs arrested in G2/M phase after injury contribute to production of profibrogenic growth factors, such as transforming growth factor beta 1 (TGF-β1) and connective tissue growth factor (CTGF)[6]. These profibrotic factors not only promote proliferation and activation of fibroblasts through paracrine effects, stimulating extracellular matrix (ECM) production and accumulation, but also induce the loss of epithelial phenotype in tubular epithelial cells via autocrine functions, indicating that epithelial cell cycle G2/M arrest is associated with kidney fibrosis[7]. Therefore, identifying the key and universal molecules involved in epithelial cell cycle G2/M arrest may provide new therapeutic clues to prevent kidney fibrosis and accelerated progression of chronic kidney disease.

[1]The Key Laboratory of Infection and Immunity of Shandong Province, Department of Pharmacology, School of Basic Medical Sciences, Shandong University, Jinan 250012, China. [2]Department of Neurosurgery, Provincial Hospital Affiliated to Shandong First Medical University, Jinan 250021, China. [3]Department of Organ Transplantation, Qilu Hospital of Shandong University, Jinan 250012, China. [4]Department of Nephrology, the First Affiliated Hospital of Zhengzhou University, Zhengzhou 450052, China. [5]Department of Pathology, School of Basic Medical Sciences, Shandong University, Jinan 250012, China. ✉e-mail: liuweimin@sdu.edu.cn; fanyi@sdu.edu.cn

Although emerging evidence has indicated the importance of epigenetic modification on the regulation of interstitial fibrosis, the biological function of individual isoforms of histone deacetylases (HDACs)-mediated histone or non-histone acetylation in the kidney are not fully elucidated[8]. HDACs are consisted of at least 18 members which can be divided into $Zn^{2+}$-dependent group (Class I: HDAC1-3 and 8; class II: HDAC4–7, 9, and 10; class IV: HDAC11) and $NAD^+$-dependent group (Sirtuin1-7)[8]. Several members of $Zn^{2+}$-dependent HDACs were reported to be increased in the kidney from UUO mice at translational levels[9]. Recent studies further demonstrated that HDAC3 was also elevated in fibrotic kidneys incurred by AAN or adenine-fed chronic kidney disease[10,11]. Moreover, the induction of HDAC6 was also observed in the kidney from angiotensin II-infused mice[12]. These studies strongly suggest that aberrant expression of $Zn^{2+}$-dependent HDACs is involved in the transmission of signals under the condition of renal interstitial fibrosis. Notably, although a variety of pan-inhibitors of HDAC also exhibited obvious effect of anti-renal fibrosis in different experimental models, indicating that pharmacological targeting of $Zn^{2+}$-dependent HDACs may be an innovative therapeutic strategy for decelerating progression of kidney fibrosis, different cellular locations and downstream targets of HDACs indicate different roles in kidney tubulointerstitial fibrosis. In addition, the high specificity and low side effects for HDAC inhibitors remain the great challenges. Therefore, further studies are needed to continue deciphering the role of individual HDAC in different physiological and pathological situations in the kidney.

The present study was designed to explore the role of HDAC9 in kidney fibrosis induced by AAN and UUO, two independent models for kidney fibrosis. We found that HDAC9 was significantly upregulated in the fibrotic kidneys, especially in proximal tubules, and the level of HDAC9 was positively correlated with α-SMA and vimentin. Tubule-specific deletion of *HDAC9* attenuated epithelial cell cycle arrest in G2/M by inhibiting the activation of STAT1. Collectively, our results demonstrated that HDAC9 may be an attractive therapeutic target for kidney fibrosis.

## Results

### HDAC9 was significantly increased in mouse and human fibrotic kidneys

To clarify the expression patterns of $Zn^{2+}$-dependent HDACs in kidney fibrosis, we performed global gene expression profiling in the cortex of kidney from aristolochic acid nephropathy (AAN). Although our results showed that *HDAC7* was elevated in the cortex of kidney from AAN, *HDAC9* was preferentially increased in fibrotic kidneys compared to the other members of $Zn^{2+}$-dependent HDACs (Fig. 1a). By real-time qRT-PCR, Western blot (WB) and immunohistochemical (IHC) staining, the upregulation of HDAC9 was further confirmed in the cortex of kidney from 2 different mouse models induced by AAN and UUO (Fig. 1b–d). Notably, the level of HDAC9 was positively correlated with Vimentin and α-SMA staining in AAN mice (Fig. 1e, f, Supplementary Fig. S1a). To better define the tubular segment specificity of HDAC9 expression in the kidney, we performed double immunostaining for HDAC9 and various tubular markers. The following segment-specific tubular markers were used: proximal tubule, aquaporin 1 (AQP1) and lotus tetragonolobus lectin (LTL); ascending loop of Henle, Tamm-Horsfall glycoprotein (THP); distal convoluted tubule, calbindin D28k; collecting duct, dolichos biflorus agglutinin (DBA)[13,14]. Our results showed that HDAC9 was significantly increased in proximal tubules from AAN mice, but there were no obvious changes in other segments of tubule (Fig. 1g–i). In addition, the upregulation of HDAC9 was also found in another two animal models, unilateral IRI and bilateral IRI (Supplementary Fig. S1b, c). In vitro, we demonstrated that HDAC9 was significantly induced in the proximal tubular cell line, humanHK-2 cells (HK-2), with aristolochic acid (AA) or TGF-β1 treatment (Fig. 1j, Supplementary Fig. S1d). Moreover, we detected the expression of HDAC9 in renal biopsies from patients with CKD (Supplementary Table S1).

Compared with the normal kidney tissues from patients who underwent tumor nephrectomy without other renal disease, CKD samples showed significant interstitial fibrosis measured by Masson´s trichrome and Sirius Red staining (Fig. 2a). Our results further showed that HDAC9 was significantly increased in kidney from CKD patients, with upregulation of Vimentin and α-SMA (Fig. 2a). Importantly, linear regression analysis showed that the level of HDAC9 in tubule was positively correlated with Vimentin and α-SMA staining (Fig. 2b, c), suggesting that HDAC9 may play an important role in kidney fibrosis.

### *Hdac9* deficiency attenuated kidney fibrosis

*HDAC9*−/− mice were generated to evaluate the role of HDAC9 in kidney fibrosis (Supplementary Fig. S2a–c). All mice were viable and fertile. *HDAC9*−/− mice showed normal physiologic index and kidney function (Supplementary Table S2). We found that *HDAC9* knockout attenuated tubular injury and tubulointerstitial fibrosis in AAN evidenced by the results of H&E staining, Masson´s trichrome and Sirius Red staining (Supplementary Fig. S3a). Moreover, *HDAC9* deficiency significantly reduced the protein level of Collagen I, Collagen IV, Vimentin, α-SMA and TGF-β1 (Supplementary Fig. S3b, c). Based on the obvious upregulation of HDAC9 in proximal tubule areas, we generated tubule-specific *HDAC9* knockout mice using a *Cre-LoxP* recombination system to better elucidate the role of HDAC9 in renal tubular epithelial cell (Supplementary Fig. S4a). *Cdh16-Cre* mice (mice expressing Cre-recombinase under the cadherin 16 promoter, Jackson Laboratory, Stock No: 012237) were crossed with *HDAC9*fl/fl mice to generate *Cdh16-Cre/HDAC9*fl/fl mice (*Cre*+/*HDAC9*fl/fl mice), which was confirmed by tail genotyping (Supplementary Fig. S4a, b), the decrease of HDAC9 in the cortex of kidney through mRNA analysis and a significant reduction of HDAC9 in isolated tubule (Supplementary Fig. S4c, d). However, there was no obvious change of HDAC9 in isolated glomeruli from *Cre*+/*HDAC9*fl/fl mice (Supplementary Fig. S4d). Meanwhile, we further demonstrated that HDAC9 was deleted in renal tubules including proximal tubule by immunofluorescence staining (Supplementary Fig. S4e). All mice were viable and fertile. *Cre*+/*HDAC9*fl/fl mice did not show any physiologic changes, including in body weight, kidney weight, heart rate, blood pressure, serum creatine (SCr) and blood urea nitrogen (BUN) (Supplementary Table S3). Moreover, we did not find any obvious changes in kidney structure (Fig. 3a), indicating that tubule-specific deletion of *HDAC9* cannot cause the phenotype changes in mice under normal conditions. However, *HDAC9* deficiency in TECs significantly attenuated tubular injury according to the H&E staining (Fig. 3a). Compared with control group, AA-treated mice exhibited obvious deposition of extracellular matrix (ECM), but tubule-specific deletion of *HDAC9* alleviated kidney tubulointerstitial fibrosis evidenced by Masson´s trichrome and Sirius Red staining (Fig. 3a). We further confirmed the antifibrotic effect of *HDAC9* deficiency in AAN mice by the reduction of markers associated with fibrosis, such as Collagen I, Collagen IV, Vimentin and α-SMA (Fig. 3a, b). Moreover, our results showed that infiltration of macrophages (stained by F4/80, a cellular marker of macrophages) in kidney from AAN mice was significantly increased, which was partially reversed by tubule-specific deletion of *HDAC9* (Fig. 3a). *HDAC9* deficiency in TECs attenuated inflammatory responses by reducing the levels of proinflammatory mediator, such as *IL-1β*, *IL-6* and *TNFα* (Fig. 3c–e). In vitro, gene silence of *HDAC9* decreased the protein level of Vimentin and Collagen I (Fig. 3f), as well as reduced the mRNA levels of *IL-6* and *TNFα* in HK-2 cells with AA treatment (Fig. 3g–h). In addition, TGF-β1 treatment enhanced the expression of markers associated with fibrosis and proinflammatory mediators, which was also reversed by *HDAC9* knockdown in HK-2 (Supplementary Fig. S5a–c).

### HDAC9 contributed to epithelial cell cycle arrest in G2/M

Notably, knockout of *HDAC9* reduced the percentage of TECs arrested in the G2/M phase based on the expression of phospho-histoneH3 at

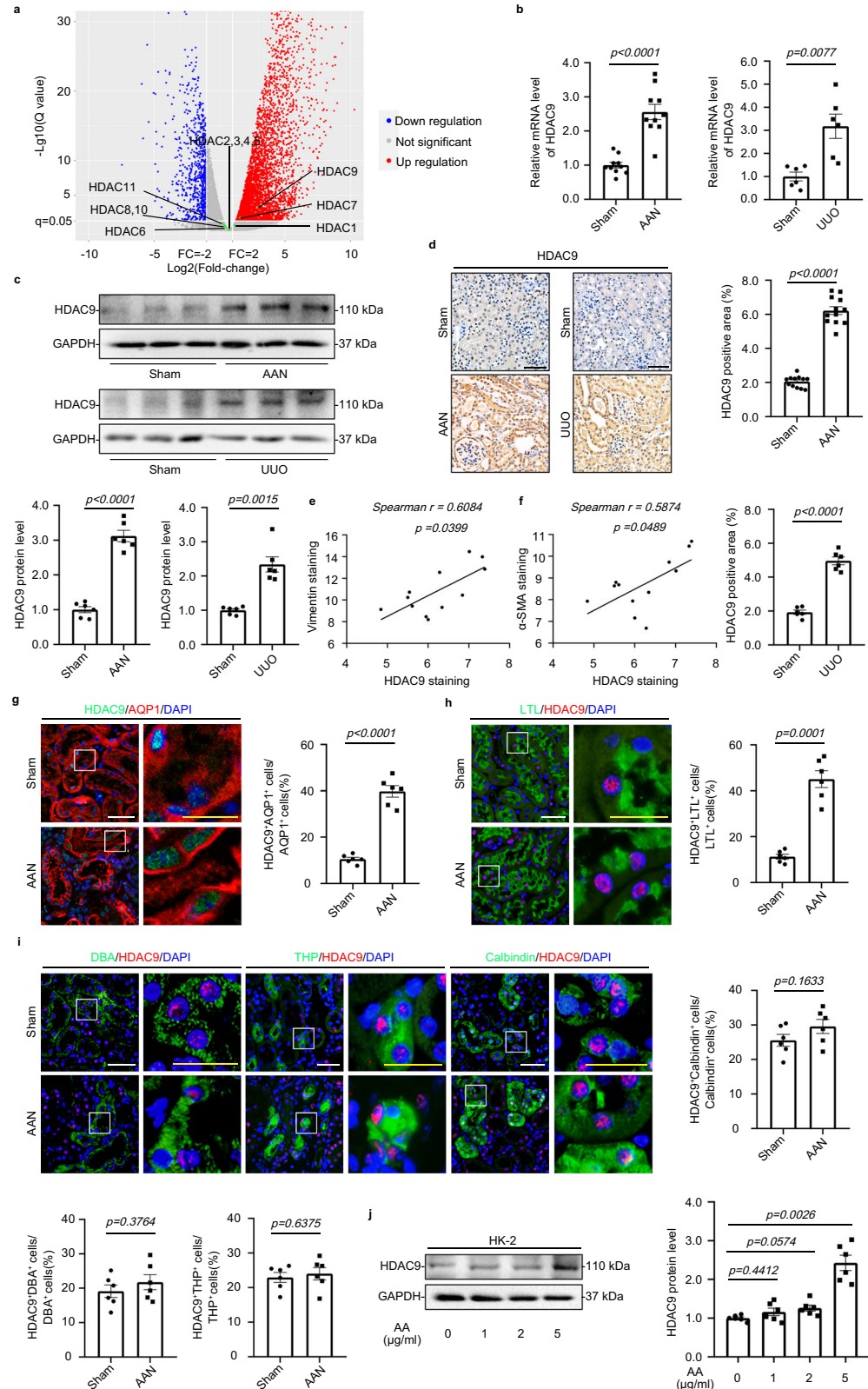

serine10 (p-H3 positive), a marker of cell cycle arrest in the G2/M phase[15], among all proliferative TECs (Ki-67 positive), which was further confirmed by the reduction in the ratio of cyclin B1/cyclin D1 and p21 expression (Supplementary Fig. S6a, b), two important molecules related to G2/M arrest[16], suggesting HDAC9 might contribute to epithelial cell G2/M arrest. To further investigate the role of HDAC9 in PTECs, we analyzed the cell cycle distribution of proximal tubular

epithelial cells by flow cytometry analysis and found that AA or TGF-β1 treatment increased the percentage of cells in G2/M phase, which was attenuated by *HDAC9* knockdown (Fig. 4a, Supplementary Fig. S7a–c). Moreover, gene silencing of *HDAC9* reduced the cyclin B1/cyclin D1 ratio and the expression of p21 (Fig. 4b). In addition, we analyzed the profibrotic mediators TGF-β1 and CTGF, two fibrogenic growth factors which can be generated from PTECs arrested in the G2/M phase and

**Fig. 1 | HDAC9 was significantly increased in mouse fibrotic kidneys, especially in proximal tubules. a** Volcano plot showing *HDACs* expression in the cortex of kidney from AAN (*n* = 4 mice for sham group, *n* = 6 mice for AAN group). Statistically significant changes are marked to indicate whether genes are expected to increase (red) or decrease (blue). **b** Relative mRNA level of *HDAC9* in the cortex of kidney from AAN (*n* = 10 mice per group) and UUO (*n* = 6 mice per group) mice. **c** Protein levels of HDAC9 in the cortex of kidney from AAN and UUO mice (*n* = 6 mice per group). **d** Photomicrographs and quantifications showing HDAC9 expression in kidney from AAN (*n* = 12 mice per group) and UUO (*n* = 6 mice per group) mice. Scale bar: black = 50 μm. Correlation between HDAC9 expression and the degree of Vimentin (**e**) or α-SMA (**f**) staining in AAN mice. (*n* = 12 mice). HDAC9 expression in proximal tubules from AAN mice (*n* = 6 mice per group). Aquaporin 1 (AQP1) (**g**) and lotus tetragonolobus lectin (LTL) (**h**) were used as a proximal tubular marker. Scale bar: white = 20 μm, yellow = 10 μm. **i** HDAC9 expression in tubules from AAN mice (*n* = 6 mice per group). Tamm-Horsfall glycoprotein (THP) was used as a marker for ascending loop of Henle; calbindin D28k was used as a marker for distal convoluted tubule; dolichos biflorus agglutinin (DBA) was used as a marker for collecting duct. Scale bar: white = 20 μm, yellow = 10 μm. **j** Protein levels of HDAC9 in human tubule epithelial cells (HK-2) with aristolochic acid (AA) treatment for 48 h. (*n* = 6 biologically independent experiments). Data are expressed as mean ± SEM (**b**, **c**, **d**, **g**, **h**, **i** and **j**). Two-tailed Student's unpaired *t* test analysis (**b**, **c**, **d**, **g**, **h** and **i**), one-way ANOVA followed by Tukey's post-test (**j**), nonparametric Spearman's correlation coefficient *r* with two-tailed *p*-value (**e** and **f**). Source data are provided as a Source Data file.

induce proliferation and activation of fibroblasts[6]. Our results showed that TGF-β1 was not only able to induce NRK-49F cell proliferation and matrix production evidenced by an increased expression of PCNA and α-SMA (Supplementary Fig. S7d, e) but also promoted the loss of epithelial phenotype (Supplementary Fig. S5a). However, *HDAC9* knockdown partially reversed upregulation of *CTGF* and TGF-β1 in HK-2 treated by AA (Supplementary Fig. S8a, b). We further examined the protein contents of TGF-β1 in supernatant of AA-treated HK-2 cells by ELISA. As shown in Fig. 4c, AA treatment increased the secretion of TGF-β1 in supernatant of HK-2, which was inhibited by *HDAC9* knockdown, indicating that HDAC9-mediated G2/M arrest in PTECs contributed to generation of fibrogenic growth factors. To explore the causal relationship between HDAC9-mediated G2/M arrest in PTECs and fibrosis, we cultured fibroblasts in vitro using the conditioned medium from AA–treated HK-2 cells and found conditioned medium from AA-treated HK-2 cells promoted activation of fibroblast, which was reversed by gene silencing of *HDAC9* in HK-2 cells (Fig. 4d, e), further suggesting that HDAC9-mediated G2/M arrest contributes to activation of fibroblasts. In vivo, tubule-specific deletion of *HDAC9* reduced the percentage of TECs arrested in the G2/M phase, and decreased the ratio of cyclin B1/cyclin D1 and p21 expression (Fig. 4f, g, Supplementary Fig. S8c). Furthermore, *HDAC9* deficiency in TECs downregulated the expression of TGF-β1 in the kidney from AAN mice (Fig. 4g). In addition, tubule-specific *HDAC9* knockout reduced the protein level of PCNA and α-SMA, especially in PDGRF-β-positive fibroblasts from fibrotic kidneys (Fig. 4h), suggesting *HDAC9* deficiency in TECs inhibited proliferation and activation of fibroblasts in AAN.

## HDAC9 promoted STAT1 activation by deacetylation

STAT1 has been reported to be deacetylated by HDACs, thus permitting its phosphorylation and reactivation, finally leading to tubulointerstitial fibrosis[17,18]. Here, we assessed the expression patterns of STATs in HK-2 cells and found that AA treatment promoted STAT1 and STAT6 phosphorylation, but reduced the phosphorylation of STAT2, STAT3 and STAT5, and had no obvious effects on STAT4 phosphorylation (Fig. 5a). Importantly, gene silencing of *HDAC9* inhibited STAT1 phosphorylation but increased STAT1 acetylation in HK-2 with AA (Fig. 5a, b). Meanwhile, *HDAC9* knockdown reduced the accumulation of p-STAT1 in nucleus of tubular epithelial cells (Fig. 5c). We further demonstrated that the interaction between HDAC9 and STAT1 was strengthened in HK-2 with AA treatment (Fig. 5d), suggesting that HDAC9 specifically activated STAT1 by reducing the acetylation and increasing the phosphorylation of STAT1 after binding. In vivo, We also found that the binding between HDAC9 and STAT1 was increased in AA treated mice, with upregulation of STAT1 phosphorylation (Fig. 5e, f). However, tubule-specific deletion of *HDAC9* reduced STAT1 phosphorylation (Fig. 5f). In addition, the increased phosphorylation of STAT1 was further confirmed in the kidney from CKD patients, with upregulation of cyclin B1(Fig. 5g). Importantly, linear regression analysis showed that the level of p-STAT1 in kidney was positively correlated with HDAC9, cyclin B1, α-SMA and Vimentin expression

(Fig. 5h–k). Furthermore, overexpression of STAT1 (Supplementary Fig. S9a) counteracted the effect of *HDAC9* knockdown in HK-2 by increasing the level of p-H3, cyclin B1 and p21, as well as contributing to the increase in the percentage of cells in G2/M (Fig. 6a–d, Supplementary Fig. S9b). Gene silence of *HDAC9* in HK-2 reduced the production of TGF-β1 and prevented proliferation and activation of fibroblasts evidenced by downregulation of α-SMA, Vimentin and PCNA, which was also abrogated by overexpression of STAT1 (Fig. 6e, f). In addition, overexpression of STAT1 elevated the expression of α-SMA and Vimentin in AA treated HK-2 with *HDAC9* knockdown (Supplementary Fig. S9c). These results indicate that STAT1 may be a key molecule linking HDAC9 to G2/M arrest in tubular epithelial cells.

## *HDAC9* deletion ameliorated tubulointerstitial fibrosis induced by UUO

We also utilized the murine model of UUO to confirm the role of HDAC9 in tubulointerstitial fibrosis. Tubule-specific deletion of *HDAC9* ameliorated tubular atrophy and tubulointerstitial fibrosis in UUO mice according to the results of H&E staining, Masson´s trichrome and Sirius Red staining (Fig. 7a), which was further confirmed by reduction of Collagen I, Collagen IV, Vimentin and α-SMA (Fig. 7a and Supplementary Fig. S10a). *HDAC9* deficiency in TECs attenuated inflammatory responses by reducing macrophage accumulation and the levels of proinflammatory mediator in kidney from UUO mice (Fig. 7a, b). We also found that *HDAC9* deficiency in TECs decreased the percentage of TECs arrested in the G2/M phase evidenced by reduction of p-H3 positive staining, cyclin B1 and p21 expression (Fig. 7c, d), suggesting *HDAC9* deficiency reduced G2/M phase arrest in TECs from UUO mice. Furthermore, *HDAC9* deficiency in TECs downregulated the expression of TGF-β1 in the kidney from UUO mice (Fig. 7e). Finally, our results showed that *HDAC9* deficiency in TECs inhibited proliferation and activation of fibroblasts in UUO measured by PCNA and α-SMA staining (Fig. 7f).

## Pharmacological inhibition with TMP195 attenuated kidney fibrosis

To test the therapeutic implication of our observations, TMP195, a selective class IIa HDAC inhibitor with high affinity for HDAC9[19], was utilized in AAN mice (Supplementary Fig. S11a). Our results showed that TMP195 alleviated tubular atrophy and tubulointerstitial fibrosis in AAN mice (Fig. 8a, Supplementary Fig. S11b). TMP195 administration decreased inflammatory responses and attenuated G2/M phase arrest in TECs (Fig. 8a–c). We further observed that the expression of p21 and TGF-β1 were reduced in the kidney from TMP195-treated AAN mice (Fig. 8d). In addition, pharmacological inhibition with TMP195 also inhibited the activation of fibroblast according to the decrease of PCNA and α-SMA, especially in PDGRF-β-positive fibroblasts in AAN mice (Fig. 8e). In vitro, TMP195 alleviated the loss of epithelial phenotype and downregulated proinflammatory mediators in HK-2 with AA treatment (Supplementary Fig. S11c–e). Moreover, TMP195 decreased the percentage of HK-2 cells in G2/M and inhibited upregulation of p21 (Fig. 8f, g,

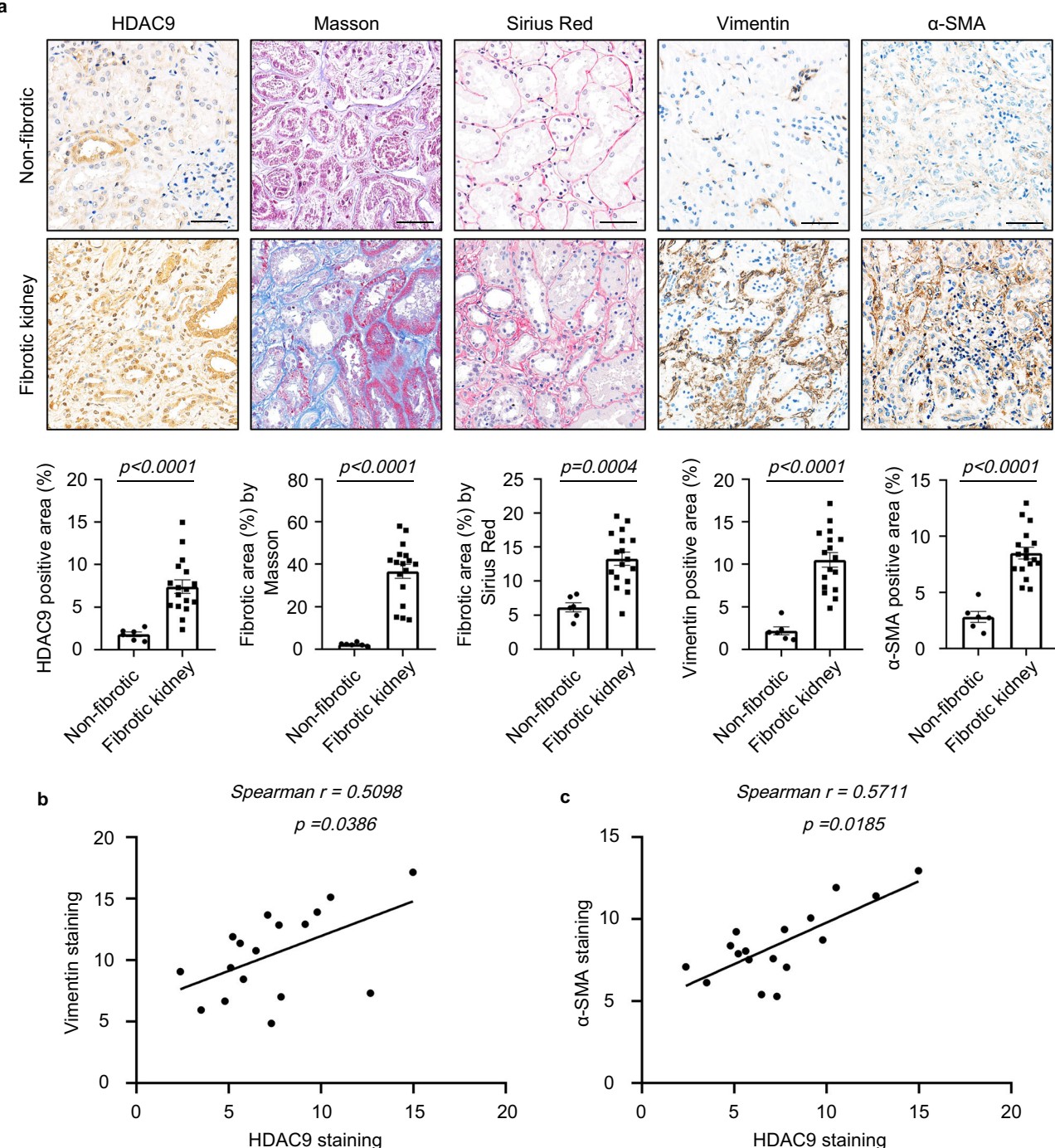

**Fig. 2 | HDAC9 was significantly increased in human fibrotic kidneys.**
**a** Photomicrographs and quantifications of HDAC9, Masson´s trichrome, Sirius Red Vimentin and α-SMA staining in kidney sections of patients with CKD. Scale bar: black = 50 μm. (*n* = 6 for normal subjects, *n* = 17 for patients with CKD).
**b** Correlation between HDAC9 expression and the degree of Vimentin staining in all subjects. (*n* = 17). **c** Correlation between HDAC9 expression and the degree of α-SMA staining in all subjects. (*n* = 17). Data are expressed as mean ± SEM (**a**). Two-tailed Student's unpaired *t* test analysis (**a**), nonparametric Spearman's correlation coefficient *r* with two-tailed *p*-value (**b** and **c**). Source data are provided as a Source Data file.

Supplementary Fig. S11f). Our results further showed that pharma-cological inhibition with TMP195 reduced the expression and secretion TGF-β1 in supernatant of HK-2 with AA (Fig. 8g, h). Finally, we demonstrated that conditioned medium from TMP195-treated HK-2 cells inhibited activation of fibroblast compared to AA-treated HK-2 conditioned medium (Fig. 8i). Additionally, TMP195 reduced the level of STAT1 phosphorylation in AAN mice, suggesting inhibi-tion of HDAC9 inhibited STAT1 phosphorylation (Supplementary Fig. S11g).

## Discussion

HDAC9, a member of class IIa HDACs, has been implicated in lipid metabolism, progression of atherosclerosis, and macrophage polar-ization and cerebral ischemia/reperfusion injury[20]. In the kidney, HDAC9 was demonstrated to contribute to podocyte injury and renal damage under diabetic conditions[21,22]. However, whether HDAC9 is a universal pathogenic factor contributing to kidney fibrosis under dif-ferent pathophysiological conditions in chronic kidney disease keeps unclear. In this study, our results showed that HDAC9 expression was

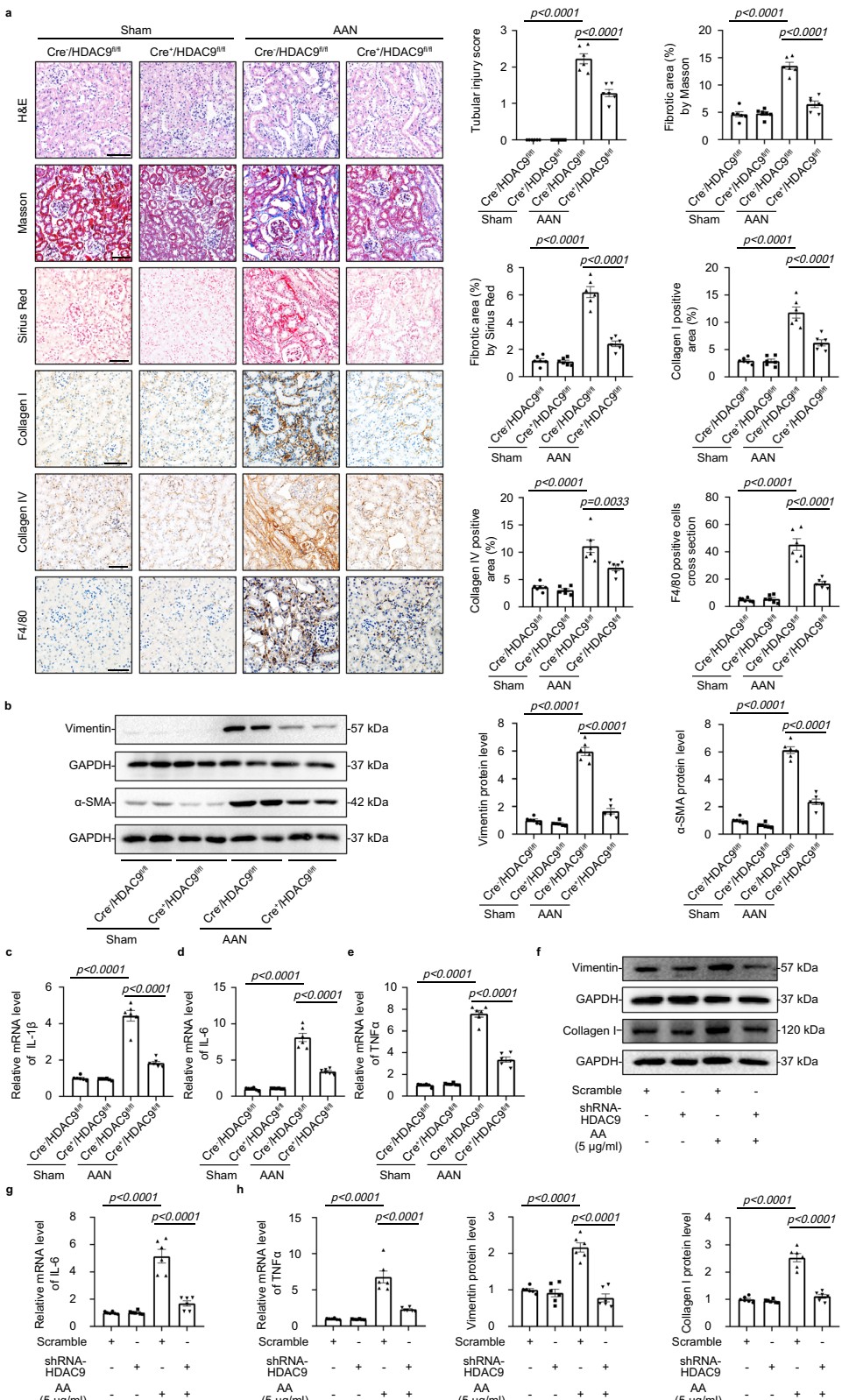

**Fig. 3 | *HDAC9* deficiency attenuated kidney fibrosis and inflammation. a** H&E staining, Masson´s trichrome staining, Sirius Red staining, collagen I staining, collagen IV staining and F4/80 staining were performed to assess the kidney injury, fibrosis and inflammation in different groups. Scale bar: black = 50 µm. (*n* = 6 mice per group). **b** Representative Western blot gel documents and summarized data showing the relative protein levels of Vimentin and α-SMA in the cortex of kidney from different groups of mice. (*n* = 6 mice per group). **c–e** Relative mRNA level of *IL-1β*, *IL-6* and *TNFα* in the cortex of kidney from AAN mice. (*n* = 6 mice per group). **f** Representative Western blot gel documents and summarized data showing the relative protein levels of Vimentin and Collagen I in HK-2 with AA treatments. (*n* = 6 biologically independent experiments). **g, h** Relative mRNA level of *IL-6* and *TNFα* in HK-2 with AA treatments. (*n* = 6 biologically independent experiments). Data are expressed as mean ± SEM (**a**–**h**). Two-way ANOVA followed by Tukey's post-test (**a**–**h**). Source data are provided as a Source Data file.

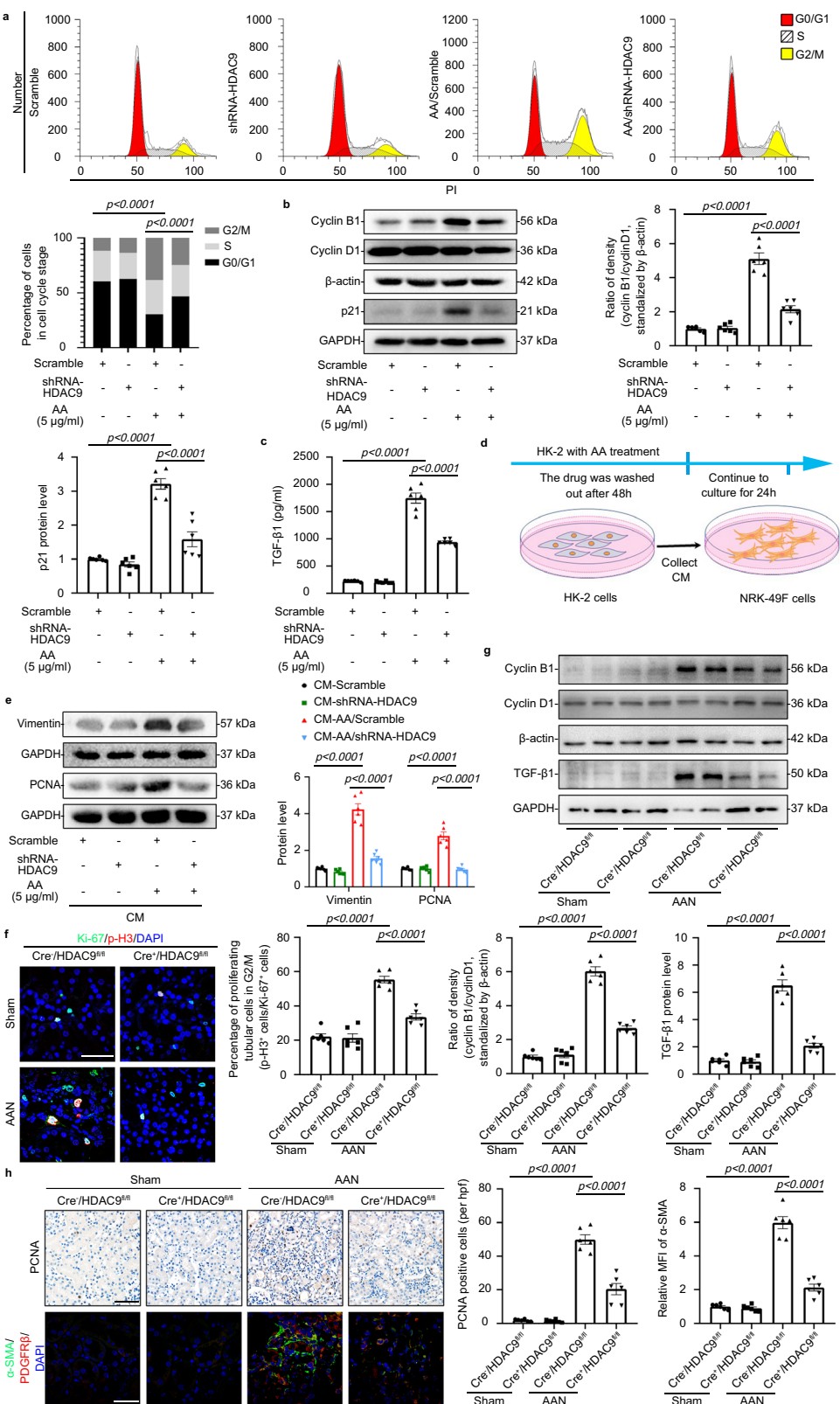

significantly increased in fibrotic kidneys, especially in proximal tubules, from different mouse models of kidney fibrosis including AAN, UUO and ischemia-reperfusion injury (IRI)-induced chronic kidney disease. Importantly, we demonstrated that a significant increase of tubule HDAC9 was positively correlated with Vimentin and α-SMA in fibrotic kidney, further suggesting that HDAC9 is a universal pathogenic factor and plays an important role in kidney fibrosis.

An increasing number of studies have demonstrated that tubulointerstitial damage is correlated better with kidney function decline than that of glomerular injury[23]. Kidney tubulointerstitial fibrosis is one of the major pathological features of chronic kidney diseases and involves multiple cell types[24–29]. Tubular epithelial cells, as the major component of the kidney, are vulnerable to various injuries and might be entered into the stage of maladaptive repair, followed by

**Fig. 4 | HDAC9 contributed to epithelial cell cycle arrest in G2/M. a** Cell cycle analysis by flow cytometry for HK-2 in different groups. ($n = 6$ biologically independent experiments). **b** Representative Western blot gel documents and summarized data showing the ratio of cyclin B1 to cyclin D1 densities standardized to β-actin and protein levels of p21 in HK-2 with AA treatment. ($n = 6$ biologically independent experiments). **c** The level of TGF-β1 in the culture supernatant from HK-2 treated with AA for 24 h. ($n = 6$ biologically independent experiments). **d** Experimental scheme for cell treatment: after 48 h treatment of HK-2 cells with aristolochic acid, the drug was washed out and the cells continued in culture for 24 h. The conditioned medium was then collected and added to serum-starved NRK-49F cells. **e** Protein levels of Vimentin and PCNA in fibroblasts treated with conditioned medium from control or G2/M-arrested HK-2 cells. ($n = 6$ biologically independent experiments). **f** Representative photomicrographs of coimmunostaining with antibodies to Ki-67 (anti−Ki-67) and p-H3 (anti−p-H3) on kidneys and the percentage of Ki-67[+] p-H3[+] cells among total Ki-67[+] tubular cells in different groups. Scale bar: white = 20 μm. ($n = 6$ mice per group). **g** Representative Western blot gel documents and summarized data showing the ratio of cyclin B1 to cyclin D1 densities standardized to β-actin in isolated tubules and protein levels of TGF-β1 in cortex of kidney from different groups. ($n = 6$ mice per group). **h** Photomicrographs and quantifications showing PCNA expression in kidney from different groups of mice (up panel). PCNA-positive cells per high power field (hpf) are counted and shown. Scale bar: black = 50 μm. Representative photomicrographs of kidney sections stained for α-SMA, PDGFRβ[+], and DAPI (down panel), as well as quantitative analysis of a-SMA staining in kidney. Scale bar: white = 20 μm. ($n = 6$ mice per group). Data are expressed as mean ± SEM (**a–c**, **e–h**). Two-way ANOVA followed by Tukey's post-test (**a–c**, **e–h**). Source data are provided as a Source Data file.

undergoing changes and functioning as fibrogenic cells, finally leading to the kidney fibrosis, indicating that tubular epithelial cells is not only as victims but also as a contributor in the progression from acute to chronic kidney disease[23]. Recently, accumulating evidence shows that cell-cycle arrest in tubular epithelial cells is an important driver of maladaptive repair and kidney fibrosis[2,6,25,30,31]. Yang et al. found that a significant increase in the proportion of PTECs arrested in the G2/M phase results in the acquisition of a pathogenic phenotype characterized by the sustained expression of profibrotic growth factors, such as TGF-β1 and CTGF[6], but reversal of the G2/M arrest attenuated kidney fibrosis by reduction of profibrotic growth factors[6]. These results indicate that targeting the G2/M checkpoints may be an attractive therapeutic strategy to prevent progressive chronic kidney disease. In this study, we demonstrated that *HDAC9* knockdown or pharmacological inhibition decreased the percentage of HK-2 cells in G2/M phase induced by AA or TGF-β1 treatment, which was further confirmed in tubule-specific deletion of *HDAC9* and TMP195-treated mice with AAN or UUO, suggesting that HDAC9 contributes to G2/M arrest in tubular epithelial cells. Checkpoints in the cell cycle are usually involved in a number of cell cycle regulatory proteins, such as cyclins, cyclin dependent kinase (CDK) and CDK inhibitors. P21, as a member of the CDK inhibitor family, inhibits the kinase activity of CDK1 (also known as CDC2)−cyclin B1, then inducing cellular G2/M arrest[16]. It has been reported that p21 was upregulated in the fibrotic kidneys[30,32]. Furthermore, *p21* knockout mice were protected from kidney fibrosis compared with wild-type mice[33]. Consistently, our results showed that the expression of p21 was increased in the HK-2 treated by AA, with upregulation of cyclin B1, which was partially reversed by *HDAC9* deficiency, suggesting HDAC9 regulates G2/M arrest by activating p21/cyclin B1 signaling. We further demonstrated that HDAC9-mediated G2/M arrest promoted the production of profibrogenic growth factors in PTECs, especially for TGF-β1, which was not only able to induce NRK-49F cell proliferation and activation but also induced the loss of epithelial phenotype in HK-2. Collectively, we concluded that HDAC9-mediated G2/M cell cycle arrest contributes to fibroblast activation and loss of epithelial phenotype in PTECs by promoting the production of profibrogenic growth factors, such as TGF-β1, finally causing extracellular matrix (ECM) accumulation and kidney fibrosis. Moreover, TGF-β1 is also known to promote fibrosis by enhancing macrophage infiltration[34]. Our result showed that tubule-specific deletion of *HDAC9* or pharmacological inhibition by TMP195 alleviated inflammation in kidney fibrosis, suggesting HDAC9-mediated G2/M cell cycle arrest in PTECs could exacerbate inflammation responses by upregulating the production of TGF-β1. In addition, TMP195, a selective inhibitor of class IIa HDAC, was reported to reduce atheroprogression and confer plaque stability by inhibiting HDAC9[19]. Here, we found that TMP195 attenuated kidney fibrosis with inhibiting G2/M phase arrest in TECs, indicating HDAC9 may be an attractive therapeutic target for kidney fibrosis.

Mechanistically, HDAC9 induced G2/M arrest of PTECs and kidney fibrosis by the activation of STAT1 that is a member of the STAT family which functions as transcription factors that mediate cell proliferation, oxidative stress and apoptosis[35]. STAT1 can be deacetylated by HDACs, thus permitting its phosphorylation and reactivation[18]. Accumulating evidence has indicated that STAT1 activation (the phosphorylated form of STAT1, p-STAT1) contributes to tubulointerstitial fibrosis in various kinds of CKD[17,35–38]. Importantly, the activation of STAT1 regulates cell cycle arrest and senescence by upregulating p21[39–41]. In this study, we assessed the expression patterns of STATs in HK-2 cells and found that AA treatment promoted STAT1 and STAT6 phosphorylation, but reduced the phosphorylation of STAT2, STAT3 and STAT5, and had no obvious effects on STAT4 phosphorylation. Importantly, gene silencing of *HDAC9* selectively inhibited STAT1 phosphorylation but increased STAT1 acetylation in HK-2 with AA. Moreover, we demonstrated that the interaction between HDAC9 and STAT1 was strengthened in AAN, suggesting that HDAC9 activates STAT1 by reducing the acetylation and increasing the phosphorylation of STAT1 after binding. These results indicate that STAT1 may be a key molecule linking HDAC9 to G2/M arrest in tubular epithelial cells.

It should be noted that there are some limitations in this study should be noted. First, although we found the role of HDAC9 in regulating tubular epithelial cells G2/M arrest, further studies are required to delineate the mechanisms by which various stimuli increase HDAC9 expression under pathologic conditions. Regarding this issue, recent studies have reported that TGF-β1 stimulation induced the expression of DNMT3a in HK-2 cells[42], and DNMT3a could maintain high expression of HDAC9 in macrophages[43]. Our preliminary results showed that DNMT3a was increased in the kidney from AAN and UUO mice (Supplementary Fig. S12a) and in HK-2 cells with AA or TGF-β1 treatment (Supplementary Fig. S12b), indicating that DNMT3a might be one of key molecules in regulating HDAC9 expression. Second, accumulating evidence demonstrates the detrimental effect of HDACs in renal injury[8], such as HDAC3, HDAC7 and HDAC8, but whether the different members work synergistically to promote to kidney fibrosis remains unclear. Third, a recent study reported that proximal tubule cells showed features of a senescence-associated secretory phenotype but did not exhibit G2/M cell cycle arrest in post-IRI kidneys by single-nucleus RNA sequencing[44]. However, other researchers demonstrated that G2/M arrest in tubular epithelial cells is occurred and involved in the development of kidney fibrosis induced by AA[45,46]. Therefore, we propose that the different conclusion obtained may be due to the different interval of observation, severity of injury or animal models used. Hence, further studies are required to clarify the related regulatory mechanisms.

Collectively, our studies demonstrate that HDAC9 contributes to G2/M arrest in tubular epithelial cells by regulating the activation of STAT1, followed by promoting production of profibrotic cytokine, finally causing kidney tubulointerstitial fibrosis (Fig. 9).

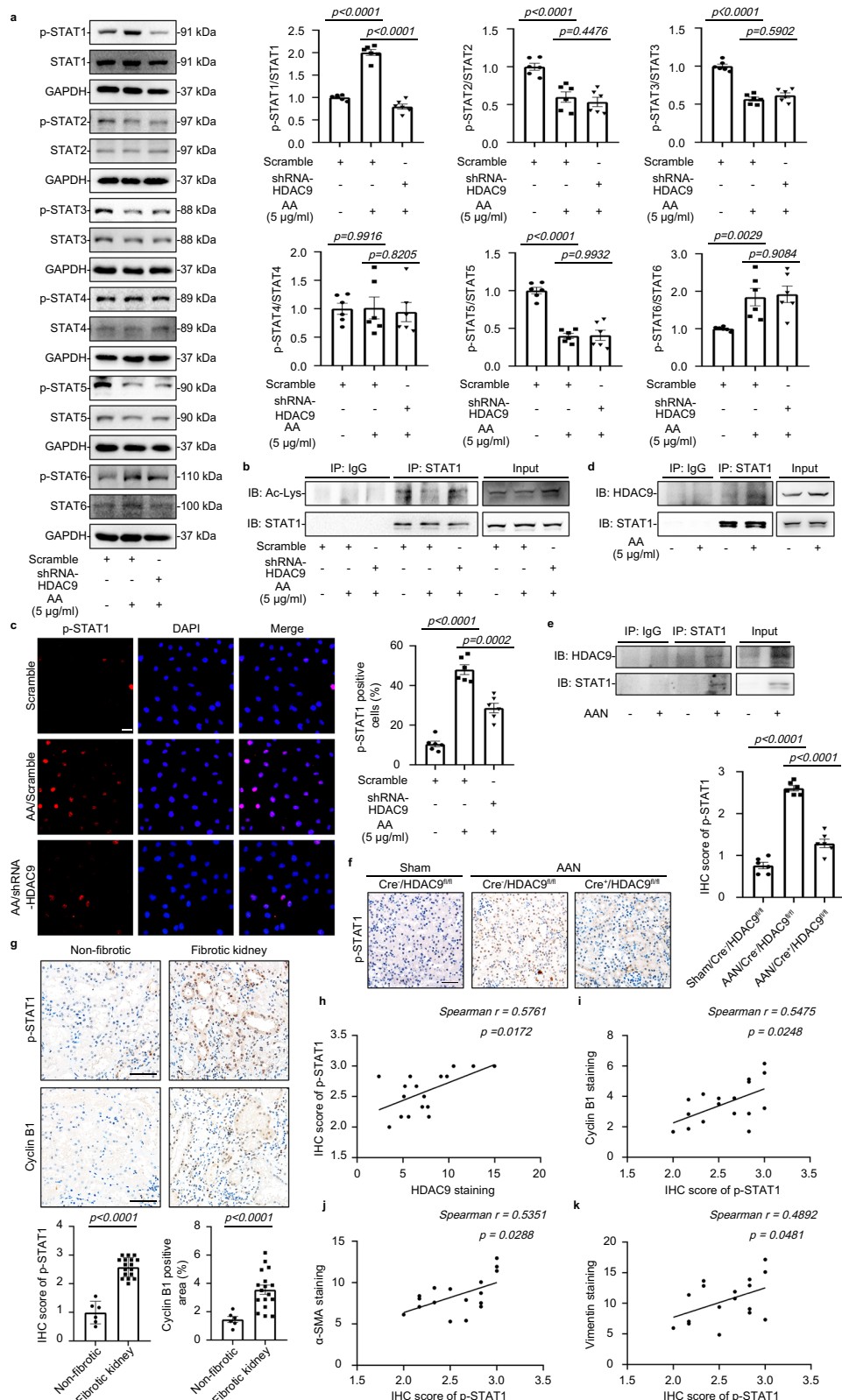

Pharmacological targeting of HDAC9 may be an effective innovative therapeutic strategy for patients with chronic kidney disease.

## Methods

### Human renal biopsy samples

Renal biopsies had been performed as part of routine clinical diagnostic investigation. We collected the human renal biopsy samples from patients with chronic kidney disease (Supplementary Table S1), including focal segmental glomerulosclerosis ($n = 9$, 7 males and 2 females), IgA nephropathy ($n = 5$, 3 males and 2 females) and diabetic nephropathy ($n = 3$, 3 males). The samples of renal biopsies were obtained from Department of Nephrology, Qilu Hospital of Shandong University, Department of Pathology, Shandong University School of Basic Medical Sciences and Department of Nephrology, the First

**Fig. 5 | HDAC9 promoted STAT1 activation by deacetylation. a** Representative Western blot gel documents and summarized data showing the phosphorylation levels of STATs in HK-2 cells. (*n* = 6 biologically independent experiments). **b** Immunoprecipitation demonstrated that *HDAC9* knockdown reduced the acetylation of STAT1. **c** Photomicrographs and quantifications showing that *HDAC9* knockdown reduced accumulation of p-STAT1 in the nucleus of AA-treated HK-2 cells. Scale bar: white = 20 μm. (n = 6 biologically independent experiments). **d.** Immunoprecipitation demonstrated that HDAC9 bound to STAT1 in HK-2. **e.** Immunoprecipitation demonstrated that HDAC9 bound to STAT1 in the cortex of kidney from AAN. **f.** Photomicrographs and quantifications showing that tubule-specific deletion of *HDAC9* reduced the protein levels of p-STAT1 in kidney from

AAN. Scale bar: black = 50 μm. (*n* = 6 mice per group). **g** Photomicrographs and quantifications of p-STAT1 and cyclinB1 in kidney from normal subjects and patients with CKD, Scale bar: black = 50 μm. (*n* = 6 for normal subjects, *n* = 17 for patients with CKD). **h**–**k** Correlation analysis between p-STAT1 and HDAC9, cyclinB1, a-SMA and Vimentin expression in all subjects. (n = 17). **b** and **d** were repeated three times independently with similar results; **e** was repeated five times independently with similar results. Data are expressed as mean ± SEM (**a**, **c**, **f** and **g**). Two-way ANOVA followed by Tukey's post-test (**a**, **c** and **f**), Two-tailed Student's unpaired *t* test analysis (**g**), nonparametric Spearman's correlation coefficient *r* with two-tailed *p*-value (**h**–**k**). Source data are provided as a Source Data file.

Affiliated Hospital of Zhengzhou University. The chronic kidney disease patients did not start dialysis therapy at the time of kidney biopsy. Normal control samples were obtained from healthy kidney poles of individuals who underwent tumor nephrectomies or renal cystectomy without other kidney diseases. The investigations were conducted in accordance with the principles of the Declaration of Helsinki and were approved by the Research Ethics Committee of Shandong University (Document No. ECSBMSSDU2018-1-051). The written, informed consent to participate was obtained from all study participants (or their parents/legal guardians). All the human study participants did not receive compensation.

## Mouse studies
All experimental protocols for animal studies were conducted in accordance with the National Institutes of Health Guide for the Care and Use of Laboratory Animals and were approved by the Institutional Animal Care and Use Committee of School of Basic Medical Sciences, Shandong University (Document No. ECSBMSSDU2018-2-091). All mice (3–5 mice per cage) were housed under SPF conditions (12 h light/dark cycle, 24 °C and 40–60% humidity) with ad libitum access to water and standard laboratory chow diet (Beijing KEAOXIELI feed company, Beijing, China). Water and cages were autoclaved. Cages with standard corncob bedding were changed three times a week. For all of the in vivo experiments, littermate control mice were used. All of our experimental animals were kept under barrier conditions under constant veterinary supervision and did not display signs of distress or pathological changes that warranted veterinary intervention. Different groups were allocated in a randomized manner and investigators were blinded to the allocation of different groups when doing surgeries and doing outcome evaluations. The number of the mice used for the experiments is indicated for each experiment in the figure legends. An established mouse model of renal IRI was performed[47]. Briefly, Male C57BL/6 mice, aged 8 weeks (24–28 g), were anesthetized with an intraperitoneal injection of pentobarbital sodium (30 mg/kg body weight) and maintained on a heat pad during surgery. A midline abdominal incision was made and bilateral renal pedicles were clipped for 32 min at 37.5 °C (bilateral IRI, BIRI) or only the left kidney for 35 min at 37 °C (unilateral IRI, UIRI). Sham operations were performed with exposure of both kidneys but without induction of ischemia. After surgery, the mice were maintained under SPF conditions (12 h light/dark cycle, 24 °C and 40–60% humidity) with ad libitum access to water and standard laboratory chow diet. The mice were euthanized and the kidney tissue samples were harvested for histopathological analysis after 4 weeks. The UUO model (Male C57BL/6 mice, aged 8 weeks, 24–28 g) was generated by ligation of the left ureter[6]. After surgery, the mice were maintained under SPF conditions (12 h light/dark cycle, 24 °C and 40–60% humidity) with ad libitum access to water and standard laboratory chow diet. After 7 days of ureteral obstruction, the mice were euthanized and the kidney tissue samples were harvested for histopathological analysis. For the aristolochic acid nephropathy, male C57BL/6 mice, aged 8 weeks (24–28 g), were used. The animal model was induced by a one-time intraperitoneal injection of aristolochic acid (5 mg/kg body weight, A5512, Sigma-Aldrich) in

PBS. The normal control mice were administered the same amount of PBS. After administration, the mice were maintained under SPF conditions (12 h light/dark cycle, 24 °C and 40–60% humidity) with ad libitum access to water and standard laboratory chow diet. The mice were euthanized and the kidney tissue samples were harvested for histopathological analysis after 28 days.

## Generation of global *HDAC9* knockout mice
Global *HDAC9* knockout (*HDAC9*^+/−) mice (C57BL/6JSmoc-*HDAC9*^em1Smoc, Shanghai Model Organisms Center, Inc., Shanghai, China) were purchased from Shanghai Model Organisms. Mouse genotyping was performed using genomic DNA isolated from mouse tails by PCR. The specific primers in this study are listed in Supplemental Table S4. Wide type: only a 513 bp band; homozygous (*HDAC9*^−/−): only a 696 bp band; heterozygous (*HDAC9*^+/−): both bands.

## Generation of tubule-specific *HDAC9* knockout mice
Floxed *HDAC9* mice(C57BL/6JSmoc-*HDAC9*^em1(flox)Smoc, Shanghai Model Organisms Center, Inc., Shanghai, China) were hybridized with transgenic mice expressing Cre-recombinase under the cadherin 16 promoter (B6.Cg–Tg(Cdh16-cre)91Igr/J, Jackson Laboratory) to generated tubule-specific *HDAC9* knockout mice (*Cdh16-Cre*^+/ *HDAC9*^fl/fl; *Cre*^+/*HDAC9*^fl/fl). Age-matched mice without Cre (*Cdh16-Cre*^− / *HDAC9*^fl/fl; *Cre*^−/*HDAC9*^fl/fl) were used as controls. Mouse genotyping was performed using genomic DNA isolated from mouse tails by PCR. The specific primers in this study are listed in Supplementary Table S4. Flox genotyping produced 546 bp and 516 bp fragments for the mutant and wild type respectively. Wide type: only a 516 bp band; homozygous (*HDAC9*^fl/fl): only a 546 bp band; heterozygous (*HDAC9*^fl/+): both bands. A 420 bp band was detected in Cre positive (*Cre*^+), but there is no band in Cre negative (*Cre*^−).

## Pharmacological inhibition of HDAC9
Male C57BL/6 mice, aged 8 weeks (24–28 g), were purchased from Beijing Vital River Laboratory Animal Technology Co., Ltd. Eight-week-old male mice were randomly assigned to following four groups: (1) control solvent injection; (2) TMP195 (50 mg/kg daily intraperitoneal injection from the first day to the last day during 4 weeks, dissolved in corn oil containing 5% DMSO, S8502, Selleck, USA); (3) renal fibrotic mice: AAN (5 mg/kg one-time intraperitoneal injection); and (4) TMP195 interventions of the fibrotic mice. Each group included at least six mice. In vitro, HK-2 cells were serum-starved for 12 h, then TMP195 (5 μM, dissolved in DMSO) was added 30 min prior to the addition of aristolochic acid (5 μg/ml, dissolved in DMSO). Cells were harvested after 48 h of treatment.

## RNA extraction and real-time RT-PCR
TRIzol reagent (Invitrogen, Carlsbad, CA) was used to extract total RNA from the tissues or cells. The UltraSYBR Mixture (CWBIO, Beijing, China) was used to perform Real-time quantitative RT-PCR (qRT-PCR). Bio-Rad iCycler system (Bio-Rad, Hercules, CA, USA) with Bio-Rad CFX Manager 2.1 software (Bio-Rad, California, USA) was used to analyze

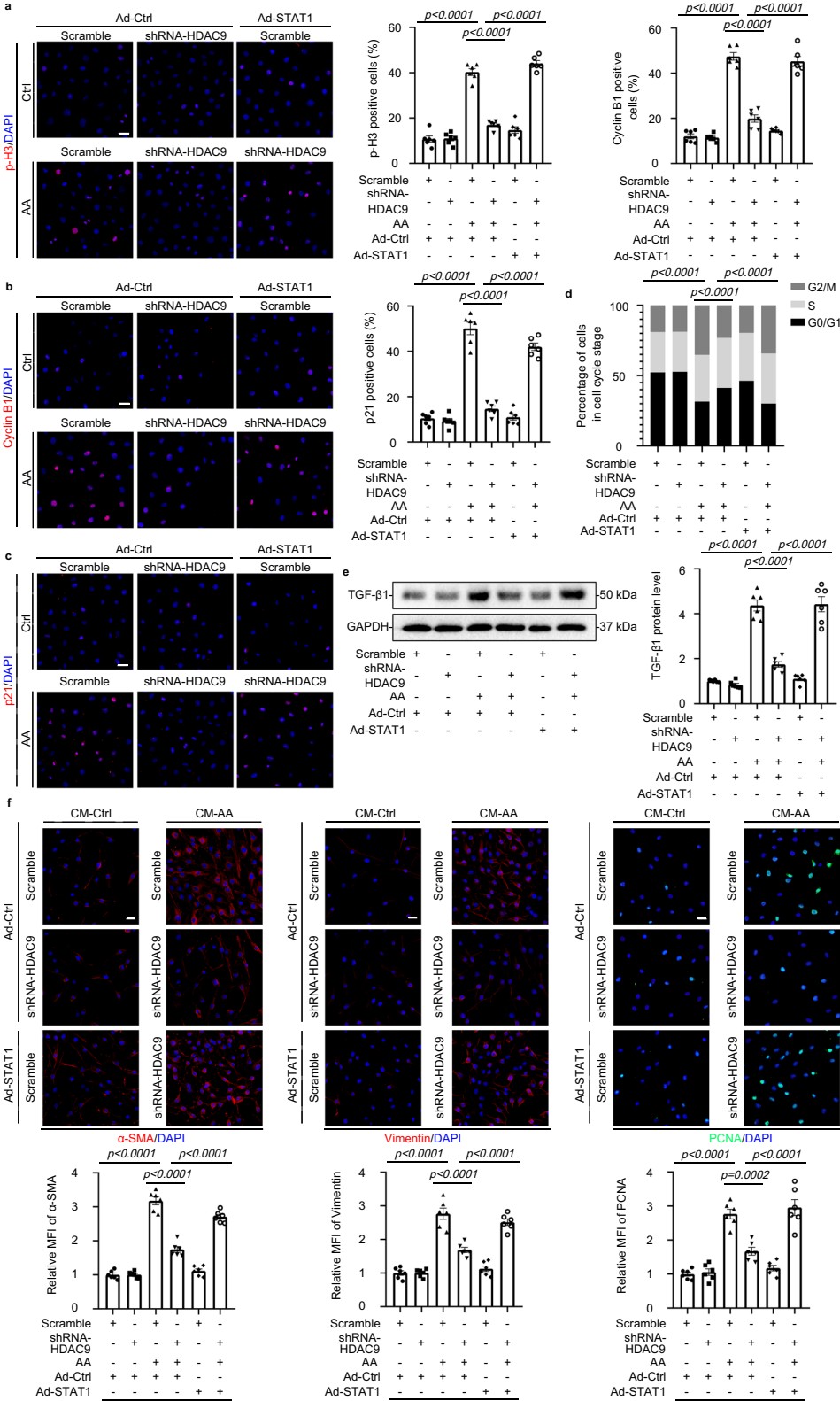

**Fig. 6 | STAT1 was involved in HDAC9-mediated G2/M arrest.**
**a**–**c** Photomicrographs and quantifications showing that overexpression of STAT1 counteracted the effect of *HDAC9* knockdown in HK-2 by increasing the level of p-H3, cyclin B1 and p21. Scale bar: white = 20 μm. (*n* = 6 biologically independent experiments). **d** Cell cycle analysis by flow cytometry for HK-2 in different groups. (*n* = 6 biologically independent experiments). **e** Representative Western blot gel documents and summarized data showing the relative protein levels of TGF-β1 in

HK-2 with different treatments. (*n* = 6 biologically independent experiments). **f** Photomicrographs and quantifications showing the relative protein levels of α-SMA, Vimentin and PCNA in fibroblasts treated with conditioned medium from HK-2 cells with different treatments. Scale bar: white = 25 μm. (*n* = 6 biologically independent experiments). HK-2 cells were stimulated with AA (5 μg/ml) for 48 h. Data are expressed as mean ± SEM (**a**–**f**). Two-way ANOVA followed by Tukey's post-test (**a**–**f**). Source data are provided as a Source Data file.

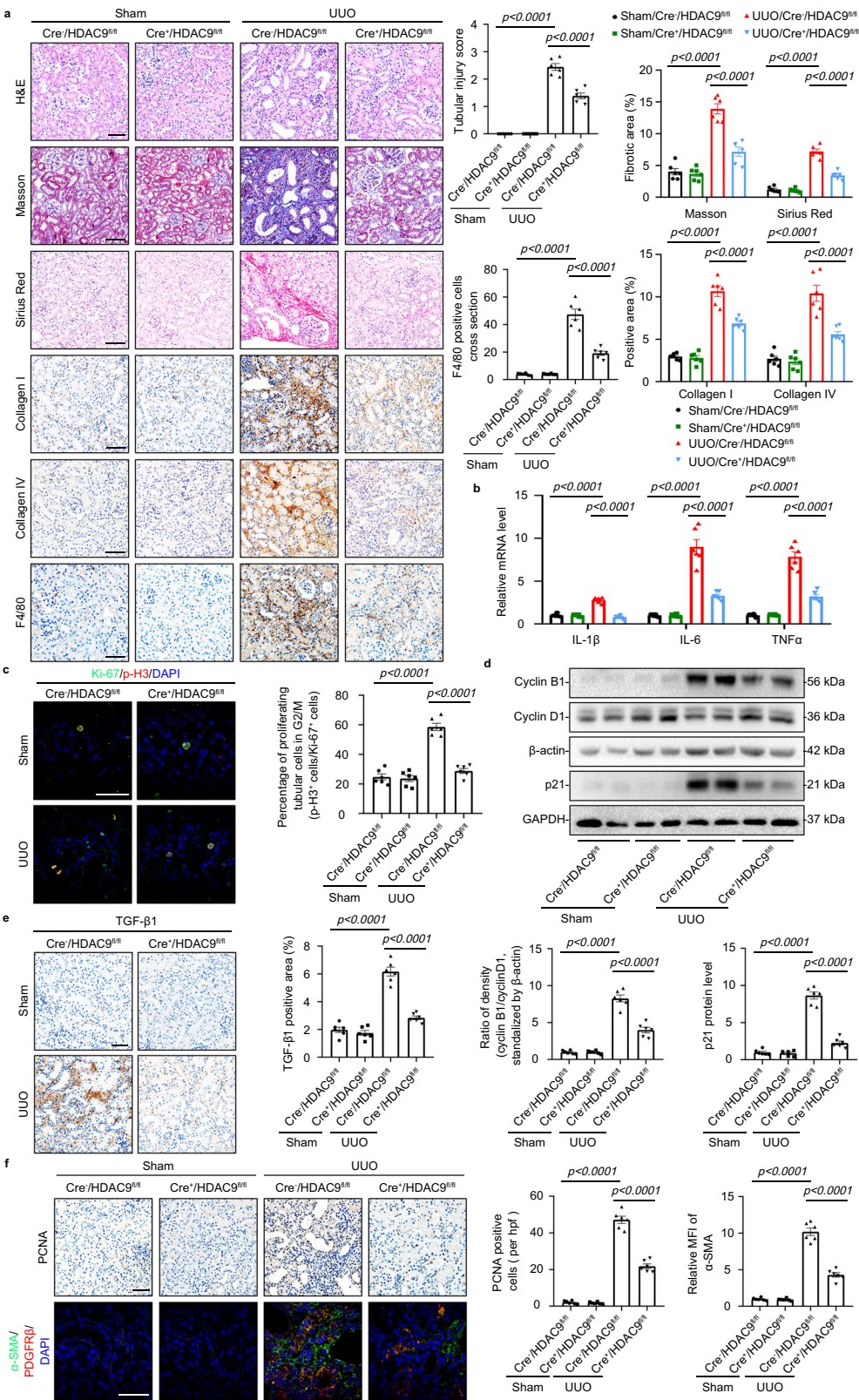

the mRNA levels for target genes. Levels of the housekeeping gene β-actin were used as an internal control. The specific primers for target genes in this study were listed in Supplementary Table S4.

## Cell culture and treatments

Human tubule epithelial cells (HK-2 cells) and normal rat kidney fibroblast cells (NRK-49F cells) were obtained from American Type Culture Collection (ATCC). HK-2 cells were cultured in Dulbecco's Modified Eagle's Medium (DMEM) supplemented with 5% fetal bovine serum (FBS) and penicillin/streptomycin. NRK-49F cells were cultured in DMEM containing 10% FBS. In vitro, HK-2 cells were serum-starved for 12 h and then treated with 5 μg/ml aristolochic acid for 48 h. In the experiment of fibroblasts culture by using the conditioned medium, the drug was washed out after the HK-2 cells were treated by AA for 48 h.

**Fig. 7 | *HDAC9* deletion ameliorated tubulointerstitial fibrosis induced by UUO.**
**a** H&E staining, Masson´s trichrome staining and Sirius Red staining were per-
formed to assess the kidney injury and fibrosis. Photomicrographs and quantifi-
cations of Collagen I and Collagen IV staining were performed to assess the kidney
fibrosis. F4/80 staining was performed to assess the kidney inflammation in dif-
ferent groups. Scale bar: black = 50 μm. (*n* = 6 mice per group). **b** Relative mRNA
level of *IL-1β*, *IL-6* and *TNFα* in the cortex of kidney from UUO mice. (*n* = 6 mice per
group). **c** Coimmunostaining with antibodies to Ki-67and p-H3 on day-7 kidneys
from different groups of mice. Scale bar: white = 20 μm. (*n* = 6 mice per group).
**d** Representative Western blot gel documents and summarized data showing the
ratio of cyclin B1 to cyclin D1 densities standardized to β-actin and the relative

protein levels of p21 in isolated tubules from different groups. (*n* = 6 mice per
group). **e** Photomicrographs and quantifications showing the expression of TGF-β1
in kidney from different groups of mice. Scale bar: black = 50 μm. (*n* = 6 mice per
group). **f** Photomicrographs and quantifications showing the expression of PCNA in
kidney from different groups of mice (up panel); PCNA-positive cells per high
power field (hpf) are counted and shown. Scale bar: black = 50 μm. Representative
photomicrographs of kidney sections stained for α-SMA, PDGFRβ⁺, and DAPI (down
panel), as well as quantitative analysis of a-SMA staining in the kidneys. Scale bar:
white = 20 μm. (*n* = 6 mice per group). Data are expressed as mean ± SEM (**a**–**f**).
Two-way ANOVA followed by Tukey's post-test (**a**–**f**). Source data are provided as a
Source Data file.

Then, HK-2 cells were cultured for 24 h and the conditioned medium
was collected to add to the serum-starved NRK-49F cells. Knockdown of
*HDAC9* by a *HDAC9*-lentivirus transfection (vectors GV115 harboring a
short-hairpin RNA sequence targeting *HDAC9*) and overexpression of
STAT1 by a *STAT1*-adenovirus transfection (vectors GV314 harboring
*STAT1*) were produced by GeneChem Co., Ltd (Shanghai, China).

**Cellular DNA flow cytometric analysis**
HK-2 cells were harvested and fixed at 4 °C overnight with 70% ethanol.
According to the manufacturer's protocol (KeyGEN BioTECH, Jiangsu,
China), the fixed cells were incubated with RNase A and PI for
30–60 min at room temperature (25 °C) in the dark after washing and
analyzed by flow cytometry (CytoFLEX, Beckman Coulter, CytExpert
2.4 version). ModFit LT 5.0 software (Verity Software House, Topsham,
ME, USA) was used to analyze cell cycle distribution.

**Western blot analysis**
In brief, tissues or cell pellets were resuspended in RIPA buffer con-
taining 1× phosphatase inhibitor 1 and 2, 1× protease inhibitor cocktail
and 1×PMSF (Phenylmethylsulfonyl fluoride), followed by incubating
in ice for at least 10 min to ensure proper cell lysis. Tissue lysates and
cells lysates were centrifuged at 4 °C for 15 min with 12,000 × *g*, then
the soluble supernatant was collected. Protein estimation was carried
out using BCA protein assay. Proteins were separated by SDS-PAGE and
transferred onto PVDF membranes. Antibodies used in this study are
summarized in Supplementary Table S5. To document the loading
controls, the membrane was reprobed with a primary antibody against
housekeeping protein GAPDH or ACTIN. Image J 1.45 software
(National Institutes of Health, Bethesda, USA) was used to perform
quantitation. Band intensity normalized to an appropriate loading
control (the housekeeping gene, GAPDH or ACTIN) and relative
abundance was presented. The primary antibodies used were as fol-
lows: anti-human/mouse HDAC9 (ORIGENE, Cat#TA324378, 1:500 for
Western blot;), anti-human Fibronectin (ProteinTech Group,
Cat#15613-1-AP, 1:1000 for Western blot), anti-human/mouse/Rat
Vimentin (Cell Signaling Technology, Cat# 5741, 1:1000 for Western
blot), anti-human/mouse/Rat Alpha-smooth muscle (Abcam, Cat#
ab124964, 1:2000 for Western blot;), anti-mouse/Rat PCNA (Pro-
teinTech Group, Cat#T10205-2-AP, 1:1000 for Western blot;), anti-
human/Rat Collagen I (Affinity, Cat# AF7001, 1:1000 for Western blot),
anti-human/mouse Cyclin B1 (ProteinTech Group, Cat# 55004-1-AP,
1:1000 for Western blot), anti-human/mouse Cyclin D1 (Abcam, Cat#
ab16663, 1:1000 for Western blot), anti-human/mouse TGF beta 1
(Abcam, Cat# ab179695, 1:2000 for Western blot), anti-human/mouse
STAT1 (phospho S727) (Abcam, Cat# ab109461, 1:1000 for Western
blot;), anti-human Acetylated-Lysine antibody (Cell Signaling Tech-
nology, Cat# 9441, 1:1000 for Western blot), anti-human STAT1 (Cell
Signaling Technology, Cat# 9176, 10 μg for IP assay; 1:1000 for Western
blot), anti-human/mouse STAT1 (Abcam, Cat# ab234400, 1:1000 for
Western blot), anti- mouse STAT1 (Abcam, Cat# ab155933, 10 μg for IP
assay; 1:1000 for Western blot), anti-human/mouse p21 (Abcam, Cat#
ab109199, 1:1000 for Western blot;), anti-human/mouse HDAC3
(ABclonal, Cat# A2139, 1:1000 for Western blot), anti-human/mouse

HDAC7 (Abcam, Cat# ab166911, 1:1000 for Western blot), anti-human/
mouse HDAC8 (ABclonal, Cat# A5829, 1:1000 for Western blot), anti-
human/mouse DNMT3a (Cell Signaling Technology, Cat# 3598, 1:1000
for Western blot), anti-human STAT2 (ABclonal, Cat# A3588, 1:1000 for
Western blot), anti-human STAT3 (ABclonal, Cat# A1192, 1:1000 for
Western blot), anti-human STAT4 (ABclonal, Cat# A4523, 1:1000
for Western blot), anti-human STAT5 (ABclonal, Cat# A5029, 1:1000
for Western blot), anti-human STAT6 (ABclonal, Cat# A19120,
1:1000 for Western blot), anti-human STAT2 (phospho Y690) (Abcam,
Cat# ab191601, 1:1000 for Western blot), anti-human STAT3 (phospho
Tyr705) (Cell Signaling Technology, Cat# 9145, 1:1000 for Western
blot), anti-human STAT4 (phospho Y690) (Santa Cruz Biotechnolog,
Cat# sc-28296, 1:500 for Western blot), anti-human STAT5 (phospho
Tyr694) (Cell Signaling Technology, Cat# 4322, 1:500 for Western
blot), anti-human STAT6 (phospho Y641) (ABclonal, Cat# AP0456,
1:500 for Western blot), anti-human/mouse/Rat GAPDH (Abways
Technology, Cat# AB0037, 1:10000 for Western blot), anti-human/
mouse β-actin (Abways Technology, Cat# AB0035, 1:10000 for Wes-
tern blot), anti-Mouse (G3A1) mAb IgG1 Isotype Control (Cell Signaling
Technology, Cat# 5415, 10 μg for IP assay), Mouse anti-rabbit IgG
(Conformation Specific) (L27A9) mAb (HRP Conjugate) (Cell Signaling
Technology, Cat# 5127, 1:2000 for Western blot).

**Immunoprecipitation assay**
STAT1 antibody (10 μg) was incubated with 30 μl Protein A&G magnetic
beads (Selleck, Huston, TX) for 1 h at room temperature with constant
rotation. The magnetic beads and the sample lysates were incubated
overnight at 4 °C. The immunoprecipitated proteins were eluted and
detected by western blot using HDAC9 and aceyl lysine antibodies.

**Histological analysis of renal tissues**
Tissues were transferred to 4% paraformaldehyde (PFA) and fixed by
leaving tissues at 4 °C overnight, then embedded in paraffin and cross-
sectioned (4 μm) for histology examination. H&E, Masson´s trichrome
and Sirius Red staining were performed according to manufactures'
instructions (Solarbio, Beijing, China). At least six randomly chosen
fields per human subject or ten randomly chosen fields per mice within
each section were photographed with Olympus BX53 (Olympus,
Tokyo, Japan) microscope at 20× or 40× magnification with cellSens
software (Olympus, Tokyo, Japan). In H&E analysis[48], renal injury was
scored according to the following system: 0 = no injury, 1 = 1%–20% of
area, 2 = 21%–50% of area, 3 = 51%–75% of area, and 4 ≥ 75%. Tubular
injury was defined as tubular sloughing, cast formation, dilatation,
degeneration, atrophy, or tubulitis. Quantification of collagen content
after Masson´s trichrome or Sirius Red staining were performed by
analyzing the % of staining area in randomly selected fields (×40) using
the Image J 1.45 software (National Institutes of Health, Bethesda, USA).
Data are expressed as positive stained area vs. total analyzed area. All
samples were examined in a blind manner.

**Immunofluorescence staining**
Tissues were transferred to 4% PFA and fixed by leaving tissues at 4 °C
overnight, then embedded in paraffin and cross-sectioned (4 μm).

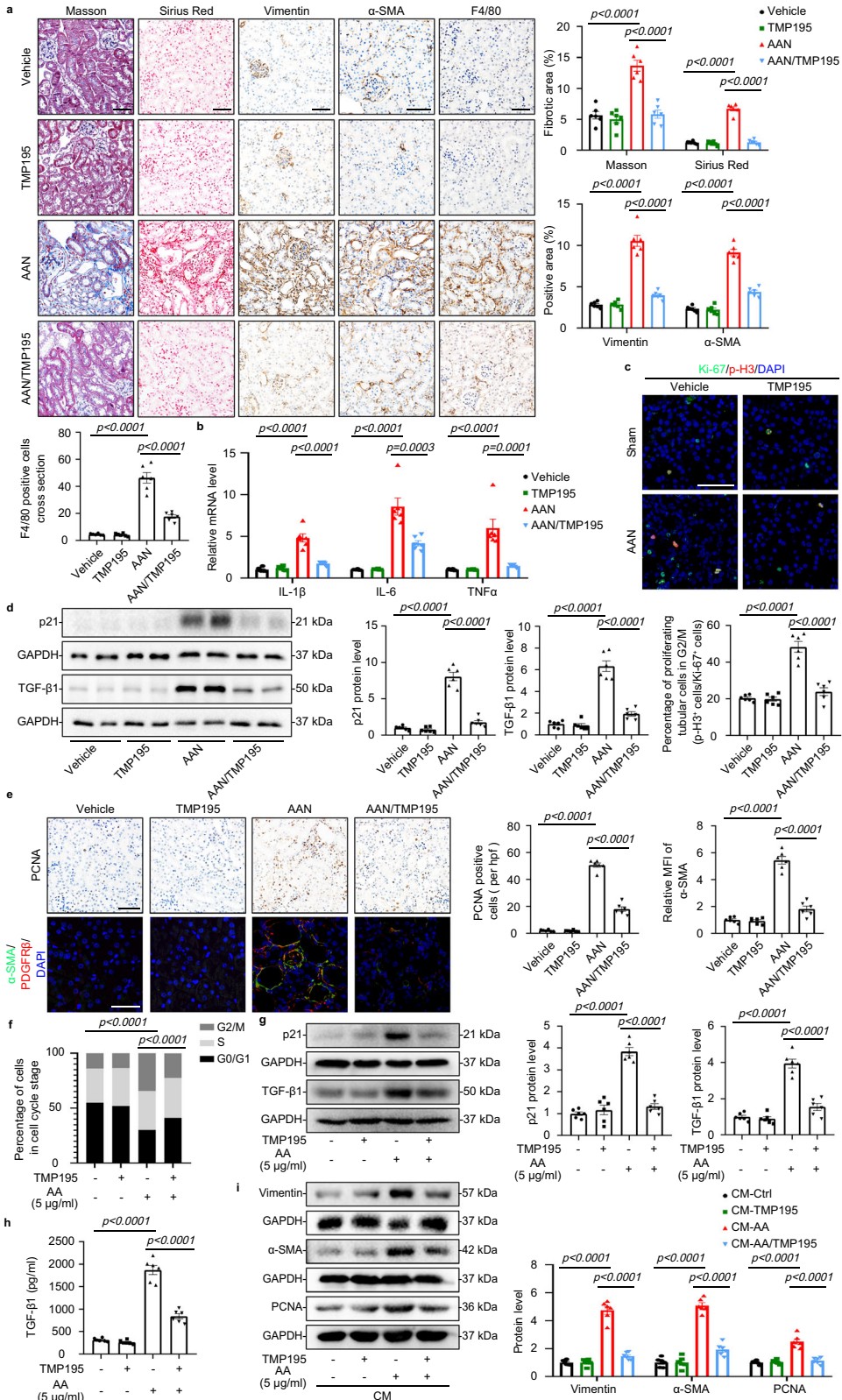

Sections were incubated with different primary antibodies and secondary Alexa 488 or 594 conjugated antibody (Abcam). DAPI (Roche, Mannheim, Germany) was used to stain Nuclei. Antibodies used in this study are summarized in Supplementary Table S5. At least ten randomly chosen fields per mice within each section were photographed with Olympus BX53 (Olympus, Tokyo, Japan) microscope at 20× or 40× magnification with cellSens software (Olympus, Tokyo, Japan) or

LSM880 laser scanning confocal microscope (ZEISS, Oberkochen, Germany) system with a Plan-Apochromat 63×/1.4 objective and ZEN 2.3 software (ZEISS, Oberkochen, Germany). The images were quantified in positive area per chosen field using the Image J 1.45 software (National Institutes of Health, Bethesda, USA). In vitro, the images were obtained by Olympus BX53 microscope or LSM880 laser scanning confocal microscope system at 20× or 40× magnification.

**Fig. 8 | Pharmacological inhibition with TMP195 attenuated kidney fibrosis.** **a** H&E staining, Masson´s trichrome staining and Sirius Red staining were performed to assess kidney injury and fibrosis. Photomicrographs and quantifications of Vimentin and α-SMA staining were performed to assess kidney fibrosis. F4/80 staining was performed to assess kidney inflammation in different groups. Scale bar: black = 50 μm. (*n* = 6 mice per group). **b** Relative mRNA level of *IL-1β*, *IL-6* and *TNFα* in the cortex of kidney from AAN mice. (*n* = 6 mice per group). **c** Representative photomicrographs of coimmunostaining with antibodies to Ki-67 (anti−Ki-67) and p-H3 (anti−p-H3) on kidneys and the percentage of Ki-67⁺ p-H3⁺ cells among total Ki-67⁺ tubular cells in different groups. Scale bar: white = 20 μm. (*n* = 6 mice per group). **d** Protein levels of p21 and TGF-β1 in the cortex of kidney from different groups. (*n* = 6 mice per group). **e** Photomicrographs and quantifications showing the expression of PCNA in kidney from different groups of mice

(up panel); PCNA-positive cells per high power field (hpf) are counted and shown. Scale bar: black = 50 μm. Representative photomicrographs of kidney sections stained for α-SMA, PDGFRβ⁺, and DAPI (down panel). Quantitative analysis of a-SMA staining in the kidney was performed. Scale bar: white = 20 μm. (*n* = 6 mice per group). **f** Cell cycle analysis by flow cytometry for HK-2 in different groups. (*n* = 6 biologically independent experiments). **g** Protein levels of p21 and TGF-β1 in HK-2 with AA treatment. (*n* = 6 biologically independent experiments). **h** The level of TGF-β1 in the culture supernatant from HK-2 treated with AA for 24 h. (*n* = 6 biologically independent experiments). **i** Protein levels of Vimentin, α-SMA and PCNA in fibroblasts treated with conditioned medium from HK-2 cells with different treatments. (*n* = 6 biologically independent experiments). Data are expressed as mean ± SEM (**a**–**i**). Two-way ANOVA followed by Tukey's post-test (**a**–**i**). Source data are provided as a Source Data file.

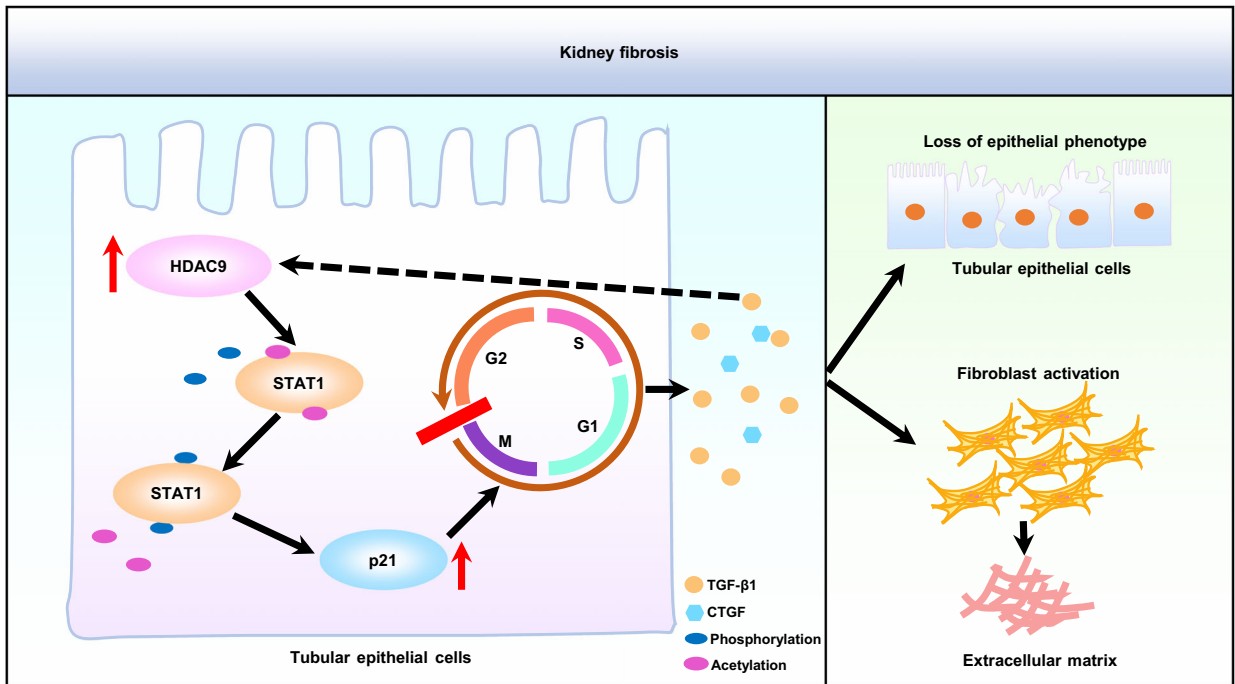

**Fig. 9 | Schematic depicting HDAC9-mediated epithelial cell cycle arrest in G2/M contributes to kidney fibrosis.** In pathological condition, HDAC9 contributes to G2/M arrest in tubular epithelial cells by regulating the activation of STAT1, followed by inducing production of profibrotic cytokine, such as TGF-β1, which promoted the loss of epithelial phenotype in tubular epithelial cells and activation of fibroblasts evidenced by upregulation of profibrotic genes, finally causing kidney tubulointerstitial fibrosis. HDAC9 histone deacetylase 9, STAT1 signal transducer and activator of transcription 1, p21 cyclin-dependent kinase inhibitor p21, G2 G2 phase of the cell cycle, M M phase of the cell cycle, G1 G1 phase of the cell cycle, S S phase of the cell cycle, TGF-β1 transforming growth factor β1, CTGF connective tissue growth factor.

Quantification was performed through analyzing at least 50 cells per group in one experiment by Image J 1.45 software (National Institutes of Health, Bethesda, USA). All samples were examined in a blind manner.

### Isolation of renal tubules
The isolation of renal tubules was performed according to a modified method for glomeruli isolation[49]. Briefly, mice were anesthetized and the surgical procedures were performed, then kidneys were minced into small pieces and pressed through a 100 μm cell strainer after digestion. The glomeruli containing Dynabeads ($8 \times 10^7$ beads/mouse, Invitrogen, M-450 Tosylactivated) were gathered by a magnetic particle concentrator after washing, then the tubules in residual resuspension were also collected for protein or mRNA analysis.

### Blood pressure measurements in conscious mice
The tail-cuff system (Softron BP-2010; Softron, Tokyo, Japan) with BPTerm10AU BP-2010 software (Softron Tokyo, Japan) was used to

measure systolic and diastolic blood pressure in mice[49]. After 5 days of training, measurements of blood pressure in mice were performed at day time (2:00 p.m. to 5:00 p.m.) and taken three times consecutively for each mouse. The blood pressure was represented by the averaged data at that time point.

### RNA-sequencing analysis
According to the manufacturer's instruction, mouse kidney RNA was isolated using MJzol animal RNA Extraction Kit (MagBeads, Cat#T102096). TruSeq® RNA Sample Preparation Kit (Illumina, California, USA) was utilized to generate Paired-end libraries. The quantification of libraries was performed by Qubit® 2.0 Fluorometer (Thermo Fisher Scientific, Waltham, USA). Agilent 2100 bioanalyzer (Agilent Technologies, California, USA) was used to validate the libraries. The sequencing was performed by Shanghai Biotechnology Corporation using the Illumina HiSeq Xten (Illumina, California, USA). For the analysis of gene expression, the cleaned reads were

mapped to GRCm38.p4 (https://www.ebi.ac.uk/ena/browser/view/GCA_000001635.6) reference genome using Hisat2 (version:2.0.4) after filtering out rRNA reads, sequencing adapters, short-fragment reads and other low-quality reads. FPKM values for known gene models were generated by Stringtie (version:1.3.0). Differentially expressed genes were identified by edgeR and were selected using the following filter criteria: FDR ≤ 0.05 and fold-change ≥2 or ≤0.5.

## Statistical analyses

Data are expressed as mean ± SEM of at least three biological replicates. Statistical analyses were performed with GraphPad Prism (version 8.0, GraphPad Software, San Diego, CA). Kolmogorov–Smirnov test was used to assess the normality assumption of the data distribution. For normally distributed data, two-tailed Student's $t$ test was used to analyze the differences between two groups. For non-normally distributed data, Mann-Whitney rank sum test was used to analyze the differences between two groups. One-way ANOVA followed by post hoc Tukey's test was used to analyze differences between multiple groups with one variable. Two-way ANOVA followed by post hoc Tukey's test was used to compare multiple groups with more than one variable. Spearman correlation analysis was performed to assess the coefficient (r) and p value. Linear regression was performed to depict the linear relationship among variables. Spearman correlation analysis and Linear regression were performed by GraphPad Prism. All statistical details regarding p-value and n can be found in main and supplementary figures and figure legends. $p < 0.05$ was considered statistically significant. Different groups of mice were allocated in a randomized manner and investigators were blinded to the allocation of different groups when doing surgeries and doing outcome evaluations. No animals were excluded from the study based on animal well-being at the beginning of the study.

## Reporting summary

Further information on research design is available in the Nature Portfolio Reporting Summary linked to this article.

# Data availability

The authors declare that all data supporting the findings of this study are available within the article and its Supplementary information files. RNA-sequencing data sets have been deposited to Gene Expression Omnibus under accession code GSE217176. GRCm38.p4 reference genome in this study is available at the European Nucleotide Archive (https://www.ebi.ac.uk/ena/browser/view/GCA_000001635.6). Source data are provided with this paper.

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

## Acknowledgements

This study was supported by National Nature Science Foundation of China (81800645 to M.L., 91949202 to F.Y., 82090024 to F.Y., 82090021 to M.L.); National Key R&D Program of China (2020YFC2005000 to F.Y.); Shandong Provincial Natural Science Foundation, China (ZR2019BH030 to M.L., ZR2019ZD40 to F.Y.); China Postdoctoral Science Foundation (2018M640634 to M.L.); The Fundamental Research Funds of Shandong University (21510078614058 to M.L.); Future Scholar Program of Shandong University (21510089964238 to M.L.).

## Author contributions

Yang. Z., Y.Y., X.L. and M.L. conducted in vivo and in vitro experiments, performed data analysis, and helped write the manuscript. P.Z. and J.W. contributed to the experimental design and performed in vitro experiments. Z.W., J.W., and X.W. performed in vivo animal studies. F. Yang., W.T., Y.S. and Yan. Z. helped design experiments. Yang. Z., Q.X., J.Z. and J.S. analyzed human renal biopsy samples. F.Yi. and M.L. designed the experiment, interpreted the data, wrote the manuscript. All the authors approved the final version of the manuscript for publication.

## Competing interests

The authors declare no competing interests.
