## [Peer review file · Nature Communications]

REVIEWER COMMENTS

Reviewer #1 (Remarks to the Author):

In this study, the authors revealed that HDAC9 expression was significantly increased in fibrotic kidneys induced by AAN and UUO. Furthermore, the increase of HDAC9 was positively correlated with α -SMA and vimentin. Tubule-specific deletion of Hdac9 in mice attenuated kidney fibrosis by inhibiting epithelial cell cycle arrest in G2/M. Mechanistically, they proposed that HDAC9 deacetylated STAT1 and promoted its reactivation, followed by promoting upregulation of p21, finally leading to G2/M arrest of TECs. In general, this study is novel, comprehensive and well presented. It describes an important role of HDAC9 in renal fibrosis and elucidates its potential mechanism, which may provide a potential target for the treatment of chronic kidney disease. However, there are some concerns needed to be addressed to improve the manuscript.

1. As shown in figure 1a, the authors found that HDAC7 and HDAC9 were elevated in the cortex of kidney from AAN. Did HDAC7 compensate the HDAC9 deficiency in the kidney? As such, the authors should examine the expression of HDAC7 in the kidney from Cre+/HDAC9fl/fl mice.

2. In figure 1i and j, the results showed that increase of HDAC9 was positively correlated with Vimentin and α -SMA staining. It would be better to analyze the correlation between HDAC9 and markers associated with fibrosis in AAN mice.

3. To better confirm the specificity of HDAC9 knockout in renal tubules, the authors should also examine the expression of HDAC9 in glomeruli from Cre+/HDAC9fl/fl mice.

4. TGF- β 1, a classic profibrogenic growth factor, has been considered as a secretory factor, so it is essential to examine the change of TGF- β 1 in supernatant from HK-2 with different stimuli.

5. In figure 5f, it seems different for the number of cells in CM-AA/Ad-STAT1 group from PCNA staining compared to the same group from other staining. Please explain that.

6. In supplementary table S1, it would be better to provide the information about pathological diagnosis for the patients with chronic kidney disease, which is helpful for understanding the classification of chronic kidney disease.

Reviewer #2 (Remarks to the Author):

This study by Zhang et. al addressed the role of HDAC9 in in G2/M arrest of TECs in kidney fibrosis models of AAN and UUO. Tubule-specific deletion of Hdac9 attenuated epithelial cell cycle arrest in G2/M, reduced production of profibrotic cytokine and alleviated tubulointerstitial fibrosis. In vitro, gene silencing of HDAC9 inhibited fibroblasts proliferation and activation through inhibiting epithelial cell

cycle arrest in G2/M. Mechanistically, HDAC9 deacetylated STAT1 and promoted its reactivation, followed by inducing G2/M arrest of TECs, finally leading to tubulointerstitial fibrosis

Major Concerns:

1.Ksp-cadherin is expressed in all segments of the nephron, but the highest levels of expression are in the collecting ducts, loops of Henle, and distal tubules JASN July 2002, 13 (7) 1837-1846. AQP-1 is found in the basolateral and apical plasma membranes of the proximal tubules, IHC data demonstrate that HDAC9 colocalizes with AQP1 mainly. It is not clear how KSp-cre mediated deletion of HDAC9 in non-proximal nephron segments would offer protection. NO IHC data are provided to demonstrate HDAC9 deletion in PT of Cdh16-Cre/Hdac9fl/fl mice

2.Text describing the graphical abstract at the end of the manuscript is not included in the manuscript

3.The mechanisms described although important, they are already established downstream of HDAC9. It is more important to establish

a.How HDAC9 expression/activation regulate profibrotic genes?

b.What regulates HDAC9 expression in AAN and UUO

4.Does HDAC9 plays a role in inflammation in the AAN and UUO models?

5.It is already established that HDAC's including 3 and 8 participate in UUO. Can their inhibition additively prevent CKD?

6.No pharmacological inhibition studies are presented

7.Discussion is poorly written with very little discussion previous studies related to HDAC in fibrosis

8.What is the rationale for selecting STAT1 as the sole mediator of HDAC9 downstream signaling? Are other STATs involved?

Minor:

1.Manuscript needs editing, especially in the supplementary data section

2.Some of the data eg: fig 2c needs quantification

3.last paragraph of the intro should be only a 1-2 sentence brief summary of the results

Reviewer #3 (Remarks to the Author):

In the manuscript entitled “HDAC9-mediated Epithelial Cell Cycle Arrest in G2/M Cotributes to Kidney Fibrosis” submitted by Zhang et al, they reported that HDAC9 in proximal tubule epithelial cells contributed to fibrosis in kidney injury. The topic regarding cell cycle arrest in renal proximal tubule cells in response to acute kidney injury is controversial. In recent paper published in PNAS (2021 Vol. 118 No. 27 e2026684118), using single-nuclear transcriptomics, the author clearly showed the diversity of proximal tubule cell states in a dynamic response to acute kidney injury. However, they showed PTCs exhibited features of a senescence-associated secretory phenotype but did not exhibit G2/m cell cycle arrest. In addition, the effect of HDAC9 in kidney injury has been investigated (Silencing of Histone Deacetylase 9 Expression in Podocytes Attenuates Kidney Injury in Diabetic Nephropathy, PMID: 27633396; LINC00162 participates in the pathogenesis of diabetic nephropathy via modulating the miR-383/HDAC9 signalling pathway, PMID: 32677473). The current study is basically descriptive and does not provide a link between proximal tubular injury and the development of kidney fibrosis. In addition, many staining and quantitative data are not convincing.

Responses to Reviewers

We thank the reviewers for their thoughtful and constructive comments, which have guided the revision of this manuscript (**Manuscript ID NCOMMS-22-47548A**). The changes in the manuscript have been highlighted in red for easy identification. The concerns have been addressed as follows:

Reviewer #1 (Remarks to the Author):

In this study, the authors revealed that HDAC9 expression was significantly increased in fibrotic kidneys induced by AAN and UUO. Furthermore, the increase of HDAC9 was positively correlated with α -SMA and vimentin. Tubule-specific deletion of *Hdac9* in mice attenuated kidney fibrosis by inhibiting epithelial cell cycle arrest in G2/M. Mechanistically, they proposed that HDAC9 deacetylated STAT1 and promoted its reactivation, followed by promoting upregulation of p21, finally leading to G2/M arrest of TECs. In general, this study is novel, comprehensive and well presented. It describes an important role of HDAC9 in renal fibrosis and elucidates its potential mechanism, which may provide a potential target for the treatment of chronic kidney disease. However, there are some concerns needed to be addressed to improve the manuscript.

1. As shown in figure 1a, the authors found that HDAC7 and HDAC9 were elevated in the cortex of kidney from AAN. Did HDAC7 compensate the HDAC9 deficiency in the kidney? As such, the authors should examine the expression of HDAC7 in the kidney from *Cre⁺/Hdac9^{fl/fl}* mice.

Answer: Thank you for your suggestion! Actually, we evaluated the expression of HDAC7 in isolated tubules from *Cre⁺/Hdac9^{fl/fl}* mice *in vivo* or HDAC9-deficient HK-2 *in vitro* (see below). Our results showed that there was no obvious change of HDAC7 in tubules from *Cre⁺/Hdac9^{fl/fl}* mice. The similar results were also demonstrated in HDAC9-deficient HK-2. These results suggested that HDAC7 could not compensate for the HDAC9 deficiency in tubule epithelial cells. Additionally, HDAC9 is reported to have several unique effects though both HDAC7 and HDAC9 belong to class IIa HDAC. For example, HDAC9 is considered as the predominant class IIa HDAC responsible for endothelial-mesenchymal transition (EndMT) by specifically regulating key EndMT-associated molecular changes, including endothelial and mesenchymal cell markers and TGF- β pathway-associated transcription factors, further suggesting that HDAC9 is capable of regulating different signaling pathways which are independent of other members from class IIa HDAC (Lecce L, Xu Y, V'Gangula B, Chandel N, *et al.* Histone deacetylase 9 promotes endothelial-mesenchymal transition and an unfavorable atherosclerotic plaque phenotype. *J Clin Invest.* 2021 Aug 2; 131(15): e131178.).

a. Representative Western blot gel documents and summarized data showing the relative protein levels of HDAC7 in isolated tubules from *Cre⁺/Hdac9^{fl/fl}* mice. (n = 6 for each group). b. Representative Western blot gel documents and summarized data showing the relative protein levels of HDAC7 in HK-2 with *HDAC9* knockdown. (n = 6 for each group). Data are expressed as mean \pm SEM. Two-tailed Student's unpaired t test analysis (a and b).

2. In figure 1i and j, the results showed that increase of HDAC9 was positively correlated with Vimentin and α -SMA staining. It would be better to analyze the correlation between HDAC9 and markers associated with fibrosis in AAN mice.

Answer: We appreciate your constructive comments! As suggested, we performed correlation analysis to clarify the relationship between HDAC9 expression and markers associated with fibrosis in AAN mice, such as Vimentin and α -SMA. Our results showed that HDAC9 was positively correlated with Vimentin and α -SMA staining in AAN mice. These results indicated that HDAC9 was involved in the development of kidney fibrosis. **The related results were added in figure 1e and f.**

3. To better confirm the specificity of HDAC9 knockout in renal tubules, the authors should also examine the expression of HDAC9 in glomeruli from *Cre⁺/Hdac9^{fl/fl}* mice.

Answer: Thank you for your careful review. In fact, we also detect the expression of HDAC9 in glomeruli from *Cre⁺/Hdac9^{fl/fl}* mice. There is no significant change of HDAC9 expression in isolated glomeruli from *Cre⁺/Hdac9^{fl/fl}* mice, but a significant

reduction of HDAC9 was observed in tubules. In addition, the expression of HDAC9 in *Cdh16-Cre/Hdac9^{fl/fl}* mice was further evaluated by double immunostaining for HDAC9 and various tubular markers. We found that *Hdac9* was deleted in renal tubules, including proximal tubules, confirming the efficiency of *Hdac9* deletion in renal tubules. **The related results were added in Supplementary Figure S4d and e.**

4. TGF- β 1, a classic profibrogenic growth factor, has been considered as a secretory factor, so it is essential to examine the change of TGF- β 1 in supernatant from HK-2 with different stimuli.

Answer: As suggested, we examined the level of TGF- β 1 in supernatant of AA-treated HK-2 cells by ELISA. As shown in **Figure 4c**, AA treatment increased the secretion of TGF- β 1 in supernatant of HK-2, which was inhibited by *HDAC9* knockdown. In addition, pharmacological inhibition by TMP195 also reduced the secretion of TGF- β 1 in supernatant of HK-2 treated by AA. These results indicated that HDAC9 contributed to the secretion of TGF- β 1 in proximal tubular epithelial cells with AA treatment. **The related results were added in Figure 4c and Figure 8h.**

5. In figure 5f, it seems different for the number of cells in CM-AA/Ad-STAT1 group from PCNA staining compared to the same group from other staining. Please explain that.

Answer: Thanks for your careful review! We re-did these experiments and selected more representative images with high resolution. **The related results were added in Figure 6f.**

6. In supplementary table S1, it would be better to provide the information about pathological diagnosis for the patients with chronic kidney disease, which is helpful for understanding the classification of chronic kidney disease.

Answer: As suggested, we provided the related information about pathological diagnosis for the patients with chronic kidney disease in this revision. **The related information was added in Supplementary Table S1.**

Reviewer #2 (Remarks to the Author):

This study by Zhang *et al* addressed the role of HDAC9 in G2/M arrest of TECs in kidney fibrosis models of AAN and UUO. Tubule-specific deletion of *Hdac9* attenuated epithelial cell cycle arrest in G2/M, reduced production of profibrotic cytokine and alleviated tubulointerstitial fibrosis. In vitro, gene silencing of HDAC9 inhibited fibroblasts proliferation and activation through inhibiting epithelial cell cycle arrest in G2/M. Mechanistically, HDAC9 deacetylated STAT1 and promoted its reactivation, followed by inducing G2/M arrest of TECs, finally leading to tubulointerstitial fibrosis

Major Concerns:

1. Ksp-cadherin is expressed in all segments of the nephron, but the highest levels of expression are in the collecting ducts, loops of Henle, and distal tubules JASN July 2002, 13 (7) 1837-1846. AQP-1 is found in the basolateral and apical plasma membranes of the proximal tubules, IHC data demonstrate that HDAC9 colocalizes with AQP1 mainly. It is not clear how Ksp-cre mediated deletion of HDAC9 in non-proximal nephron segments would offer protection. NO IHC data are provided to demonstrate *Hdac9* deletion in PT of *Cdh16-Cre/Hdac9^{fl/fl}* mice.

Answer: Thank you very much for your professional questions. In this manuscript, we performed double immunostaining for HDAC9 and various tubular markers to better define the tubular segment specificity of HDAC9 expression in the kidney. The following segment-specific tubular markers were used: proximal tubule, aquaporin-1 (AQP1) and lotus tetragonolobus lectin (LTL); ascending loop of Henle, Tamm-Horsfall glycoprotein (THP); distal convoluted tubule, calbindin D28k; collecting duct, dolichos biflorus agglutinin (DBA). Our results showed that HDAC9 was significantly increased in proximal tubules from AAN mice, but there were no obvious changes in other segments of tubule, suggesting that HDAC9 was mainly involved in the injury of proximal tubule under AAN. **The related results were added in figure 1g-i.**

Ksp-Cre has been reported to be expressed in the renal tubules, including Bowman's capsule, proximal tubule, loop of Henle and distal tubule (Kai T, *et al*. Kidney-specific knockout of *Sav1* in the mouse promotes hyperproliferation of renal tubular epithelium through suppression of the Hippo pathway. *J Pathol*, 2016 May; 239(1): 97-108). It means that Ksp-Cre could induce *Hdac9* knockout in proximal tubules. In this revision, the expression of HDAC9 in *Cdh16-Cre/Hdac9^{fl/fl}* mice was evaluated by double immunostaining for HDAC9 and various tubular markers. We found that *Hdac9* was deleted in renal tubules, including proximal tubule. Meanwhile, our results also showed that HDAC9 was significantly reduced in isolated tubule rather than in isolated glomeruli from *Cdh16-Cre/Hdac9^{fl/fl}* mice by WB analysis. These results indicated Ksp-Cre could mediate deletion of *Hdac9* in tubules, including proximal tubules. **The related results were added in Supplementary Figure S4d and e.**

2. Text describing the graphical abstract at the end of the manuscript is not included in the manuscript.

Answer: Thank you for your careful review! We are sorry that the description of graphical abstract was missing in the manuscript. We have carefully checked and added the exact information at the end of discussion. **The exact information was added in page 17 highlighted in red.**

3. The mechanisms described although important, they are already established downstream of HDAC9. It is more important to establish

a. How HDAC9 expression/activation regulate profibrotic genes?

Answer: Thank you for your question! Recently, accumulating evidence shows that prolonged G2/M cell cycle arrest in proximal tubular epithelial cells (PTECs) is an important driver of maladaptive repair and kidney fibrosis. PTECs arrested in G2/M phase after injury may transform to a secretory phenotype and contribute to production of profibrogenic growth factors, such as transforming growth factor beta (TGF- β 1) and connective tissue growth factor (CTGF) (Yang L, *et al.* Epithelial cell cycle arrest in G2/M mediates kidney fibrosis after injury. *Nat Med.* 2010 May; 16(5): 535-543). Moreover, the profibrotic factors derived from PTECs arrested in G2/M not only promote proliferation and activation of fibroblasts through paracrine effects, stimulating extracellular matrix (ECM) production and accumulation, but also induce the loss of epithelial phenotype in tubular epithelial cells via autocrine functions (Man J Livingston, *et al.* Tubular cells produce FGF2 via autophagy after acute kidney injury leading to fibroblast activation and renal fibrosis. *Autophagy.* 2023 Jan; 19(1): 256-277). These studies highlighted the importance of profibrogenic growth factors, such as TGF- β 1, in the relationship between G2/M arrest of PTECs and kidney fibrosis.

In this study, our results showed that inhibition or tubule-specific deletion of *Hdac9* attenuated G2/M cell cycle arrest in PTECs, then reduced production of TGF- β 1. We further found that TGF- β 1 was not only able to induce NRK-49F cell proliferation and activation but also promoted the loss of epithelial phenotype in HK-2. Therefore, we proposed that HDAC9-mediated G2/M cell cycle arrest in PTECs contributed to fibroblast activation and loss of epithelial phenotype in PTECs by promoting the production of profibrogenic growth factors, such as TGF- β 1, finally causing ECM accumulation and kidney fibrosis. Mechanistically, HDAC9 activated STAT1 by reducing the acetylation and increasing the phosphorylation of STAT1. Collectively, we proposed that HDAC9 contributed to G2/M arrest in PTECs by activating STAT1, followed by inducing production of profibrotic cytokine, especially for TGF- β 1, which promoted the loss of epithelial phenotype in tubular epithelial cells and the activation of fibroblasts evidenced by upregulation of profibrotic genes, such as Collagen I, Vimentin and α -SMA. Of course, we cannot exclude other regulators that also may be

involved in HDAC9-mediated expression of profibrotic genes in kidney fibrosis. Therefore, further studies for other potential targets of HDAC9 are of great interest.

b. What regulates HDAC9 expression in AAN and UO?

Answer: As reported, TGF- β 1 stimulation induced the expression of DNMT3a in HK-2 cells, and DNMT3a could maintain high expression of HDAC9 in macrophages (Taotao Hu, *et al.* DNMT3a negatively regulates PTEN to activate the PI3K/AKT pathway to aggravate renal fibrosis. *Cell Signal.* 2022 Aug; 96: 110352; Li X, *et al.* Methyltransferase Dnmt3a upregulates HDAC9 to deacetylate the kinase TBK1 for activation of antiviral innate immunity. *Nat Immunol.* 2016 Jul; 17(7): 806-815), indicating that DNMT3a might upregulate HDAC9 in tubular epithelial cells under kidney tubulointerstitial fibrosis. Therefore, we detected the protein level of DNMT3a and found that DNMT3a was increased in the kidney from AAN and UO mice. *In vitro*, AA or TGF- β 1 treatment could also induced the expression of DNMT3a in HK-2 cells. These results suggested that DNMT3a might be one of key molecules in regulating HDAC9 expression. Hence, the related regulatory mechanisms in detail need further clarification in the future. **The related results were added in Supplementary Figure S12 and discussed in page 17 highlighted in red.**

4. Does HDAC9 play a role in inflammation in the AAN and UO models?

Answer: In this revised manuscript, our results showed that infiltration of macrophages in the kidney was significantly increased in AAN and UO mice, which were partially reversed by tubule-specific deletion of *Hdac9*. Furthermore, *Hdac9* deficiency in TECs attenuated inflammatory responses by reducing the levels of proinflammatory mediator, such as IL-1 β , IL-6 and TNF α , in AAN and UO mice. *In vitro*, HDAC9 knockdown reduced the mRNA levels of IL-6 and TNF α in HK-2 with AA or TGF- β 1 treatment. In addition, we also demonstrated that pharmacological inhibition by TMP195 reduced the infiltration of macrophages in AAN mice and downregulated the levels of proinflammatory mediator, which was further confirmed in AA-treated HK-2. Collectively, these results indicated that inhibition of HDAC9 could alleviate inflammation in kidney fibrosis. **The related results were added in figure 3a, c-e, g-h; figure 7a and b; figure 8a and b; Supplementary Figure S5b and c; Supplementary Figure S11d and e. The related discussion was also provided in page 15 highlighted in red.**

5. It is already established that HDAC's including 3 and 8 participate in UO. Can their inhibition additively prevent CKD?

Answer: Thank you for your questions. Recently, an increasing number of studies reported that HDAC3 contributed to kidney fibrosis in AAN, UO and adenine-fed chronic kidney disease (Fang Chen, *et al.* Histone deacetylase 3 aberration inhibits Klotho transcription and promotes renal fibrosis. *Cell Death Differ.* 2021 Mar; 28(3):

1001–1012; Lin W, *et al.* Klotho restoration via acetylation of Peroxisome Proliferation-Activated Receptor gamma reduces the progression of chronic kidney disease. *Kidney Int.* 2017 Sep; 92(3): 669-679). Meanwhile, HDAC8 was increased in the obstructed kidneys and inhibition of HDAC8 attenuated renal interstitial fibrosis after UUO injury (Zhang Y, *et al.* Identification of histone deacetylase 8 as a novel therapeutic target for renal fibrosis. *FASEB J.* 2020 Jun; 34(6): 7295-7310). These studies suggests that HDAC3 and HDAC8 are involved in kidney fibrosis and might be the therapeutic targets for chronic kidney disease. Therefore, we detected the expression of HDAC3 and HDAC8 in isolated tubules from *Cre⁺/Hdac9^{fl/fl}* mice *in vivo* or *HDAC9*-deficient HK-2 *in vitro* (see below). However, our results showed that the protein levels of HDAC3 and HDAC8 were not changed in isolated tubules from tubule-specific knockout of *Hdac9* mice or *HDAC9*-deficient HK-2, suggesting that the role of HDAC9 may be independent of HDAC3 and HDAC8. In addition, HDAC9 comes from class IIa HDAC, which is different from class I HDAC (HDAC1,2,3,8) at structure and cellular distribution, indicating that HDAC9 might possess some different properties and effects. Therefore, it is very possible that inhibition of HDAC3 and HDAC8 might additively prevent chronic kidney disease although this is not our major issue in the present study. Hence, further studies are needed to evaluate the relationship among different HDACs under chronic kidney disease.

a. Representative Western blot gel documents and summarized data showing the relative protein levels of HDAC3 and HDAC8 in isolated tubules from *Cre⁺/Hdac9^{fl/fl}* mice. (n = 6 for each group). **b.** Representative Western blot gel documents and summarized data showing the relative protein levels of HDAC3 and HDAC8 in HK-2

with HDAC9 knockdown. (n = 6 for each group). Data are expressed as mean ± SEM. Two-tailed Student's unpaired t test analysis (a and b).

6. No pharmacological inhibition studies are presented.

Answer: We really appreciate the reviewer's question! To test the therapeutic implication of our observations, TMP195, a selective class IIa HDAC inhibitor with high affinity for HDAC9, was utilized in AAN mice. Our results showed that TMP195 alleviated tubular atrophy and tubulointerstitial fibrosis in AAN mice. TMP195 administration decreased inflammatory responses and attenuated G2/M phase arrest in TECs. We further demonstrated that the expression of p21 and TGF- β 1 were reduced in the kidney from TMP195-treated AAN mice. In addition, pharmacological inhibition with TMP195 also inhibited activation of fibroblast according to the decrease of PCNA and α -SMA, especially in PDGRF- β -positive fibroblasts in AAN mice. *In vitro*, TMP195 alleviated the loss of epithelial phenotype and downregulated proinflammatory mediators in HK-2 with AA treatment. Moreover, TMP195 decreased the percentage of cells in G2/M and inhibited upregulation of P21. Our results further showed that pharmacological inhibition with TMP195 reduced the secretion TGF- β 1 in supernatant of HK-2 with AA. Finally, we demonstrated that conditioned medium from TMP195-treated HK-2 cells inhibited activation of fibroblast compared to AA-treated HK-2 conditioned medium. These results indicated that pharmacological inhibition of HDAC9 attenuated kidney fibrosis. **The related results were added in figure 8 and Supplementary Figure S11. The related discussion was added in page 15 highlighted in red.**

7. Discussion is poorly written with very little discussion previous studies related to HDAC in fibrosis.

Answer: Thank you very much for your suggestion! As suggested, we reorganized the discussion about HDACs in kidney fibrosis, especially in the expression patterns of Zn²⁺-dependent HDACs. The histone deacetylases, a family of enzymes that mediate lysine deacetylation of both histone and non-histone proteins, play an essential role in promoting kidney fibrosis (Wang J, *et al.* Molecular mechanisms of histone deacetylases and inhibitors in renal fibrosis progression. *Front Mol Biosci.* 2022 Sep 6; 9: 986405). Increasing evidence has shown that HDACs participate in the development of kidney fibrosis (Nie L, *et al.* Application of Histone Deacetylase Inhibitors in Renal Interstitial Fibrosis. *Kidney Dis (Basel).* 2020 Jul; 6(4): 226-235). All the members of Zn²⁺-dependent HDACs were reported to be increased in the kidney from UUO mice at translational levels (Wang J, *et al.* Molecular mechanisms of histone deacetylases and inhibitors in renal fibrosis progression. *Front Mol Biosci.* 2022 Sep 6; 9: 986405). Recent studies further demonstrated that HDAC3 was also elevated in fibrotic kidneys incurred by AAN or adenine-fed chronic kidney disease (Fang Chen, *et al.* Histone deacetylase 3 aberration inhibits Klotho transcription and promotes renal fibrosis. *Cell Death Differ.* 2021 Mar; 28(3): 1001-1012; Lin W, *et al.* Klotho restoration via

acetylation of Peroxisome Proliferation-Activated Receptor gamma reduces the progression of chronic kidney disease. *Kidney Int.* 2017 Sep; 92(3): 669-679). Moreover, the induction of HDAC6 was also observed in the kidney from angiotensin II-infused mice (Sin Young Choi, *et al.* Tubastatin A suppresses renal fibrosis via regulation of epigenetic histone modification and Smad3-dependent fibrotic genes. *Vascul Pharmacol.* 2015 Sep; 72 :130-140). These studies strongly suggest that aberrant expression of Zn²⁺-dependent HDACs is involved in the transmission of signals under the condition of renal interstitial fibrosis. Notably, although various HDAC inhibitors have been investigated for their antifibrotic effects in renal disease, the high specificity and low side effects remain the great challenges. Therefore, further studies are needed to continue deciphering the role of individual HDAC in different physiological and pathological situations. In this study, our results showed that Hdac9 expression was significantly increased in fibrotic kidneys, especially in proximal tubules, from different mouse models of kidney fibrosis including aristolochic acid nephropathy, unilateral ureter obstruction and ischemia-reperfusion injury (IRI)-induced chronic kidney disease. Importantly, we demonstrated for the first time that a significant increase of tubule HDAC9 was positively correlated with Vimentin and α -SMA in fibrotic kidney, further suggesting that HDAC9 plays an important role in kidney fibrosis. **The related information was added in page 4 highlighted in red.**

8. What is the rationale for selecting STAT1 as the sole mediator of HDAC9 downstream signaling? Are other STATs involved?

Answer: Thank you very much for your very constructive comments! In fact, we have assessed the expression patterns of STATs in AA-treated HK-2. Our results showed that AA treatment promoted STAT1 and STAT6 phosphorylation, but reduced the phosphorylation of STAT2, STAT3 and STAT5, and had no obvious effects on STAT4 phosphorylation. Importantly, gene silencing of *HDAC9* inhibited STAT1 phosphorylation but increased STAT1 acetylation in HK-2 with AA. Meanwhile, *HDAC9* knockdown reduced the accumulation of p-STAT1 in nucleus of tubular epithelial cells. We further demonstrated that the interaction between HDAC9 and STAT1 was strengthened in HK-2 with AA treatment, suggesting that HDAC9 specifically activated STAT1 by reducing the acetylation and increasing the phosphorylation of STAT1 after binding. *In vivo*, tubule-specific deletion of *Hdac9* or pharmacological inhibition by TMP195 reduced STAT1 phosphorylation in the kidney from AAN mice. Importantly, the increased phosphorylation of STAT1 was further confirmed in the kidney from CKD patients and the level of p-STAT1 was positively correlated with HDAC9 expression. In addition, overexpression of STAT1 counteracted the effect of *HDAC9* knockdown in HK-2 by contributing to increase the percentage of cells in G2/M phase and promoting the production of TGF- β 1, leading to proliferation and activation of fibroblasts. Collectively, these results suggested that STAT1 may be a key molecule linking HDAC9 to G2/M arrest in tubular epithelial cells. **The related results were added in figure 5, figure 6 and Supplementary Figure S11g.**

Minor:

1. Manuscript needs editing, especially in the supplementary data section.

Answer: As suggested, we reorganized our manuscript and performed English grammatical editing in this revision, especially in supplementary data section, with the support of the professional proofreader and native English speaker.

2. Some of the data eg: fig 2c needs quantification.

Answer: Thank you for your careful review. Actually, all the staining and Western blot have been quantified according to the methods as described by our previous studies (Zhan P, *et al.* Myeloid-derived growth factor deficiency exacerbates mitotic catastrophe of podocytes in glomerular disease. *Kidney Int.* 2022 Sep; 102(3): 546-559). As suggested, we have carefully checked and re-analyzed the data to get more accurate statistical analysis in this revision. In addition, the 'Source Data' was provided in a single Excel file with data for each figure/table according to the editor's request.

3. last paragraph of the intro should be only a 1-2 sentence brief summary of the results.

Answer: As suggested, we have rearranged the introduction and simplified the related description of results in this revision. Thank you very much for your careful review again!

Reviewer #3 (Remarks to the Author):

In the manuscript entitled “HDAC9-mediated Epithelial Cell Cycle Arrest in G2/M Contributes to Kidney Fibrosis” submitted by Zhang et al, they reported that HDAC9 in proximal tubule epithelial cells contributed to fibrosis in kidney injury.

1. The topic regarding cell cycle arrest in renal proximal tubule cells in response to acute kidney injury is controversial. In recent paper published in PNAS (2021 Vol. 118 No. 27 e2026684118), using single-nuclear transcriptomics, the author clearly showed the diversity of proximal tubule cell states in a dynamic response to acute kidney injury. However, they showed PTCs exhibited features of a senescence-associated secretory phenotype but did not exhibit G2/m cell cycle arrest.

Answer: We really appreciated your professional questions! Although cell cycle arrest in renal proximal tubule cells in response to acute kidney injury is controversial, an increasing number of studies have suggested that proximal tubular cell cycle arrest in the G2/M phase is related to maladaptive proximal tubule repair in kidney fibrosis. Based on different models of acute kidney injury in mice, Yang *et al.* demonstrated that a strong correlation between G2/M arrest in proximal tubular cells and a fibrotic outcome (Li Yang, *et al.* Epithelial cell cycle arrest in G2/M mediates kidney fibrosis after injury. *Nat Med.* 2010 May; 16(5): 535-543). Abrogating the G2/M arrest markedly reduced fibrosis and cytokine production, indicating that proximal tubular cells arrested in the G2/M after injury produce profibrogenic growth factors, followed by fibroblast proliferation and collagen production. Moreover, proximal tubular cell cycle arrest in the G2/M phase was also involved in Atg5-mediated autophagy deficiency and DNA damage response under kidney fibrosis (Huiyan Li, *et al.* Atg5-mediated autophagy deficiency in proximal tubules promotes cell cycle G2/M arrest and renal fibrosis. *Autophagy.* 2016 09; 12(9): 1472-1486; Seiji Kishi, *et al.* Proximal tubule ATR regulates DNA repair to prevent maladaptive renal injury responses. *J Clin Invest.* 2019 11 01; 129(11): 4797-4816). These studies suggest that proximal tubular cell cycle arrest in the G2/M phase plays an important role in kidney fibrosis.

A recent study reported that proximal tubule cells showed features of a senescence-associated secretory phenotype but did not exhibit G2/M cell cycle arrest in post-ischemia reperfusion injury (IRI) kidneys by single-nucleus RNA sequencing. Notably, to model moderate kidney injury as may occur in a variety of non-life-threatening clinical settings such as in the course of kidney transplant surgery, a mild-to-moderate ischemia-reperfusion injury model was established and kidneys were collected at day 7 or day 28 post-IRI in this study (Louisa M S Gerhardt, *et al.* Single-nuclear transcriptomics reveals diversity of proximal tubule cell states in a dynamic response to acute kidney injury. *Proc Natl Acad Sci U S A.* 2021 Jul 6; 118(27): e2026684118). However, Yang *et al* found that the percentage of proximal tubular epithelial cells in G2/M phase was reduced after day 7 though it was increased from day 1 and peaked at

day 5, suggesting that the moderate IRI mice just had transient cell cycle changes in proximal tubular cells (Li Yang, *et al.* Epithelial cell cycle arrest in G2/M mediates kidney fibrosis after injury. *Nat Med.* 2010 May; 16(5): 535-543). Therefore, we detected the percentage of tubular epithelial cells arrested in the G2/M phase from moderate IRI and severe IRI mice at day 7 after injury (see below). Our results showed that the percentage of tubular epithelial cells arrested in the G2/M phase was increased in the kidney from severe IRI mice but there was no significant change for the G2/M arrest of tubular epithelial cells in the kidney from moderate IRI mice (see below). In addition, Yang *et al* reported that there was an obvious and persistent higher number of cells in G2/M phase beginning from day 5 and remaining at high levels during the entire repair phase in AAN model. In this revision, our results also demonstrated a significant increase in the accumulation of tubular epithelial cells arrested in G2/M phase from AAN mice at day 28 after injury (figure 4f and figure 8c.). These results indicated that interval of observation, severity of injury and difference from animal models may be the key factors to be considered in the research of G2/M cell cycle arrest.

a. Representative photomicrographs of coimmunostaining with antibodies to Ki-67 (anti-Ki-67) and p-H3 (anti-p-H3) on kidneys and the percentage of Ki-67⁺ p-H3⁺ cells among total Ki-67⁺ tubular cells in different groups. Scale bar: 10 μ m. (n = 6 for each group). Data are expressed as mean \pm SEM. Two-tailed Student's unpaired t test analysis (a).

Importantly, Yueh-An Lu *et al* recently revealed phenotypes of proximal tubular cells in kidneys undergoing regeneration and fibrosis following AA exposure through single-nuclear RNA sequencing (snRNAseq) and found cell numbers of G2/M- and S-phase proximal tubular cells increased in the kidney after 4 weeks (Yueh-An Lu, *et al.* Single-

Nucleus RNA Sequencing Identifies New Classes of Proximal Tubular Epithelial Cells in Kidney Fibrosis. *J Am Soc Nephrol*. 2021 Oct; 32(10): 2501-2516). Moreover, Jiayun Chen *et al* further confirmed AA treatment severely hindered cell cycle progression in the proximal tubule from AAN mice evidenced by the increased G2/M phase (Jiayun Chen, *et al*. Integrated single-cell transcriptomics and proteomics reveal cellular-specific responses and microenvironment remodeling in aristolochic acid nephropathy. *JCI Insight*. 2022 Aug 22; 7(16): e157360). Therefore, these studies indicate that G2/M arrest in TECs is occurred and involved in the development of kidney fibrosis from AAN mice.

Collectively, controversy about G2/M arrest in renal proximal tubule cells might be attributed to interval of observation, severity of injury and difference from animal models. Of course, we cannot exclude other possibilities, such as a limitation of the snRNAseq approach, as the nuclear membrane breaks down in late G2 into M phase, releasing RNA. Therefore, further studies are also needed to be clarify the mechanism of G2/M arrest in proximal tubule cells. **The related discussion was added in page 17 highlighted in red.**

2. In addition, the effect of HDAC9 in kidney injury has been investigated (Silencing of Histone Deacetylase 9 Expression in Podocytes Attenuates Kidney Injury in Diabetic Nephropathy, PMID: 27633396; LINC00162 participates in the pathogenesis of diabetic nephropathy via modulating the miR-383/HDAC9 signalling pathway, PMID: 32677473). The current study is basically descriptive and does not provide a link between proximal tubular injury and the development of kidney fibrosis.

Answer: Thank you for your comments! HDAC9, a member of class IIa HDACs, has been implicated in lipid metabolism, progression of atherosclerosis, and macrophage polarization (Sadhan Das, Rama Natarajan. HDAC9: An Inflammatory Link in Atherosclerosis. *Circ Res*. 2020 Aug 28; 127(6): 824-826). In the kidney, HDAC9 was demonstrated to contribute to podocyte injury and renal damage under diabetic conditions (Fan W, *et al*. LINC00162 participates in the pathogenesis of diabetic nephropathy via modulating the miR-383/HDAC9 signaling pathway. *Artif Cells Nanomed Biotechnol*. 2020 Dec; 48(1): 1047-1054; Liu F, *et al*. Silencing of Histone Deacetylase 9 Expression in Podocytes Attenuates Kidney Injury in Diabetic Nephropathy. *Sci Rep*. 2016 Sep 16; 6: 33676). However, whether HDAC9 is a universal pathogenic factor contributing to kidney fibrosis under different pathophysiological conditions in chronic kidney disease keeps unclear. In the present study, HDAC9 was significantly upregulated in fibrotic kidneys, especially in proximal tubules, from different mouse models of kidney fibrosis including AAN, UO and IRI-induced chronic kidney disease. Importantly, we demonstrated for the first time that a significant increase of tubule HDAC9 was positively correlated with Vimentin and α -SMA in fibrotic kidney, further suggesting that HDAC9 may be a universal pathogenic factor and play an important role in kidney fibrosis.

Regarding to the link between proximal tubular injury and the development of kidney fibrosis, accumulating evidence shows that prolonged G2/M cell cycle arrest in proximal tubular epithelial cells (PTECs) is an important driver of maladaptive repair and kidney fibrosis. PTECs arrested in G2/M phase after injury may transform to a secretory phenotype and contribute to production of profibrogenic growth factors, such as transforming growth factor beta 1 (TGF- β 1) and connective tissue growth factor (CTGF) (Yang L, *et al.* Epithelial cell cycle arrest in G2/M mediates kidney fibrosis after injury. *Nat Med.* 2010 May; 16(5): 535-543). These profibrotic factors not only promote proliferation and activation of fibroblasts through paracrine effects, stimulating extracellular matrix (ECM) production and accumulation, but also induce the loss of epithelial phenotype in tubular epithelial cells via autocrine functions (Man J Livingston, *et al.* Tubular cells produce FGF2 via autophagy after acute kidney injury leading to fibroblast activation and renal fibrosis. *Autophagy.* 2023 Jan; 19(1): 256-277). Hence, these studies highlighted the importance of profibrogenic growth factors, such as TGF- β 1, in the relationship between G2/M arrest of PTECs and kidney fibrosis. In this study, our results showed that inhibition or knockdown of Hdac9 attenuated G2/M cell cycle arrest in PTECs, then reduced production of TGF- β 1. We further found that TGF- β 1 was not only able to induce NRK-49F cell proliferation and activation but also promoted the loss of epithelial phenotype in HK-2. Therefore, we proposed that HDAC9-mediated G2/M cell cycle arrest in PTECs contributed to fibroblast activation and loss of epithelial phenotype in tubular epithelial cells by promoting the production of profibrogenic growth factors, such as TGF- β 1, finally causing ECM accumulation and kidney fibrosis.

3. In addition, many staining and quantitative data are not convincing.

Answer: Thank you for your careful review. We have carefully checked and re-analyzed the results to get more statistical analysis. Meanwhile, images with high resolution were provided in this revision. In addition, the ‘Source Data’ was also provided in a single excel file with data for each figure/table according to the editor’s request.

REVIEWERS' COMMENTS

Reviewer #1 (Remarks to the Author):

My concerns have been fully addressed.

Reviewer #2 (Remarks to the Author):

Authors have addressed my concerns and added data and description to the manuscript

Reviewer #3 (Remarks to the Author):

The authors have satisfactorily answered questions.

Responses to Reviewers

We thank the reviewers for their valuable comments and support on this research work, which have guided the revision of this manuscript (**Manuscript ID NCOMMS-22-47548B**). The concerns have been addressed as follows:

Reviewer #1 (Remarks to the Author):

My concerns have been fully addressed.

Answer: Thank you for your review!

Reviewer #2 (Remarks to the Author):

Authors have addressed my concerns and added data and description to the manuscript

Answer: Thank you very much for your valuable comments and support on this research work.

Reviewer #3 (Remarks to the Author):

The authors have satisfactorily answered questions.

Answer: We thank the reviewer for positive comments!